# REGULARIZATION CAN MAKE DIFFUSION MODELS MORE EFFICIENT

## ABSTRACT

Diffusion models are one of the key architectures of generative AI. Their main drawback, however, is the computational costs. This study indicates that the concept of sparsity, well known especially in statistics, can provide a pathway to more efficient diffusion pipelines. Our mathematical guarantees prove that sparsity can reduce the input dimension's influence on the computational complexity to that of a much smaller intrinsic dimension of the data. Our empirical findings confirm that inducing sparsity can indeed lead to better samples at a lower cost.

## 1 INTRODUCTION

Diffusion models are probabilistic generative models that generate new data similar to those they are trained on (Song & Ermon, 2019; Ho et al., 2020; Song et al., 2021b). These models have recently gained significant attention due to their impressive performance at image generation, video synthesis, text-to-image translation, and molecular design (Dhariwal & Nichol, 2021; Ho et al., 2022; Ramesh et al., 2022; Xu et al., 2022).

A diffusion generative model is based on two stochastic processes:

1. A forward process $\boldsymbol{x}_0 \rightarrow \boldsymbol{x}_1 \rightarrow \cdots \rightarrow \boldsymbol{x}_T$ that starts from a sample $\boldsymbol{x}_0 \in \mathbb{R}^d$ from a target data distribution $q_0$ ($\boldsymbol{x}_0 \sim q_0$) and then diffuses this sample in $T$ steps into pure noise $\boldsymbol{x}_T \in \mathbb{R}^d$ ($\boldsymbol{x}_T \sim \mathcal{N}(\mathbf{0}_d, \mathbb{I}_d)$).

2. A reverse process $\boldsymbol{y}_T \rightarrow \boldsymbol{y}_{T-1} \rightarrow \cdots \rightarrow \boldsymbol{y}_0$ that starts from pure noise $\boldsymbol{y}_T \in \mathbb{R}^d$ ($\boldsymbol{y}_T \sim \mathcal{N}(\mathbf{0}_d, \mathbb{I}_d)$) and then converts this noise in $T$ steps into a new sample $\boldsymbol{y}_0 \in \mathbb{R}^d$ ($\boldsymbol{y}_0 \sim p_0$) that is similar in distribution to the target sample $\boldsymbol{x}_0 \in \mathbb{R}^d$.

Making the data noisy is easy. Therefore, fitting a good reverse process is the key to successful diffusion modeling.

The three predominant formulations of diffusion models are denoising diffusion probabilistic models (DDPM) (Ho et al., 2020), score-based generative models (SGM) (Song & Ermon, 2019), and score-based stochastic differential equations (SDE) (Song et al., 2021b;a). DDPM include two Markov chains: a forward process that transforms data into noise, and a reverse process that recovers the data from the noise. The objective is to train a function (usually a deep neural network) for denoising the data over time. Sample generation then takes random Gaussian noise through the trained denoising function. SGM, which is the setting adopted in this paper, also perturb data with a sequence of Gaussian noise but then try to estimate the score functions, the gradient of the log probability density, for the noisy data. Sampling combines the trained scores with score-based sampling approaches like Langevin dynamics. While DDPM and SGM focus on discrete time steps, Score SDEs consider infinitely many time steps or unbounded noise levels. In Score SDEs, the desired score functions are solutions of stochastic differential equations. Once the desired score functions are trained, sampling can be reached using stochastic or ordinary differential equations.

Much research efforts are geared toward non-asymptotic rates of convergence, particularly in the number of steps $T$ needed to achieve a desired level of reconstruction accuracy. Typical measures of accuracy are Kullback–Leibler divergence, total variation, and Wasserstein distance between the true distribution $\mathbb{Q}_0$ and the approximated counterpart $P_0$. For example, one tries to ensure $\mathrm{TV}(\mathbb{Q}_0, P_0) \leq \tau$ for a fixed error level $\tau \in (0, \infty)$, where $\mathrm{TV}(\mathbb{Q}_0, P_0) := \sup_{A \subset \mathbb{R}^d} |\mathbb{Q}_0(A) - P_0(A)|$ is called the

total variation (van de Geer, 2000). The many very recent papers on this topic highlight the large interest in this topic (Block et al., 2020; De Bortoli et al., 2021; De Bortoli, 2022; Lee et al., 2022; Chen et al., 2023c;a; Li et al., 2024b; Chen et al., 2023e; Liang et al., 2024b). Results like Block et al. (2020, Theorem 13) provide rates of convergence for diffusion models in terms of Wasserstein distance employing Langevin dynamics, but they suffer from the curse of dimensionality in that the rates depend exponentially on the dimensions of the data $d$, that is, the number of input features. Improved convergence rates in terms of $d$ are proposed by Chen et al. (2023c); Li et al. (2024b), showing polynomial growth in $d$. Recently Liang et al. (2024b) proposed a new Hessian-based accelerated sampler for the stochastic diffusion processes. They achieve accelerated rate for the total variation convergence for DDPMs of the order $d^{1.5}/\tau$ for any target distributions having finite variance and assuming a uniform bound over the accuracy of the estimated score function (see our Section 5 for an overview of recent works). While these results are a major step forward, they involve a strong dependence on the dimensionality of the data, which is problematic as images, text, and so forth are typically high dimensional. The key question is whether improving these rates with respect to $d$ is possible at all.

**Contribution** This paper aims to enhance the efficiency of diffusion models through the incorporation of regularization techniques commonly used in high-dimensional statistics Lederer (2022).

The contributions of this work are as follows:

- We theoretically demonstrate that $\ell_1$-regularization can enhance the convergence rates of diffusion models to the order of $s^2/\tau$, where $s \ll d$, compared to the standard order of $d^2/\tau$ (Theorem 1).

- We validate our theoretical findings through simulations on image datasets (Section 6 and Appendix).

- We additionally demonstrate that $\ell_1$-regularization can make sampling more balanced and avoid oversmoothing (Section 6.3).

Thus, our research is a step forward in the whole field's journey of improving our understanding of diffusion models and of making diffusion modeling more efficient.

**Paper outline** Section 2 introduces score matching and the discrete-time diffusion process. Section 3 presents our proposed estimator along with the main results (Theorem 1). Section 4 includes some technical results and Section 5 provides an overview of related work. We support our theoretical findings with numerical observations over image datasets in Section 6. Finally, we conclude the paper in Section 7. Additional simulations, technical results, and detailed proofs are provided in the Appendix.

## 2 PRELIMINARIES OF SCORE MATCHING AND DISCRETE-TIME DIFFUSION PROCESS

In this section, we provide a brief introduction to score matching and discrete-time diffusion process.

**Notations** For a vector $\boldsymbol{z} \in \mathbb{R}^d$, we use the notation $\|\boldsymbol{z}\|_1 := \sum_{i=1}^d |z_i|$, $\|\boldsymbol{z}\|^2 := \sum_{i=1}^d (z_i)^2$, $\|\boldsymbol{z}\|_\infty := \sup_{i \in \{1,\ldots,d\}} |z_i|$, and $\|\boldsymbol{z}\|_0 := \sum_{i=1}^d \mathbf{1}(z_i \neq 0)$.

### 2.1 SCORE MATCHING

Assume a dataset $\mathcal{D}_n := \{\boldsymbol{x}^1, \ldots, \boldsymbol{x}^n\}$ of $n$ training data samples $\boldsymbol{x}^i \in \mathbb{R}^d$ with an unknown target distribution $q_0$ ($\boldsymbol{x}^i \sim q_0$ for $i \in \{1, \ldots, n\}$). The goal of probabilistic generative modeling is to use the dataset $\mathcal{D}_n$ to learn a model that can sample from $q_0$. The score of a probability density $q(\boldsymbol{x})$, the gradient of the log-density with respect to $\boldsymbol{x}$ denoted as $\nabla_{\boldsymbol{x}} \log q(\boldsymbol{x})$, are the key components for generating new samples from $q$. The score network $\boldsymbol{s}_\Theta : \mathbb{R}^d \to \mathbb{R}^d$ is then a neural network parameterized by $\Theta \in \mathcal{B}$, which will be trained to approximate the unknown score $\nabla_{\boldsymbol{x}} \log q_0(\boldsymbol{x})$. The corresponding objective functions for learning scores in SGMs (Song & Ermon, 2019) is then

based on Hyvärinen & Dayan (2005); Hyvärinen (2007)

$$\Theta^* \in \arg\min_{\Theta \in \mathcal{B}} \mathbb{E}_{q_0(\boldsymbol{x})} \big[ \| \boldsymbol{s}_\Theta(\boldsymbol{x}) - \nabla_{\boldsymbol{x}} \log q_0(\boldsymbol{x}) \|^2 \big] , \tag{1}$$

which yields the parameters of a neural network $\boldsymbol{s}_{\Theta^*}(\boldsymbol{x})$ that approximates the unknown score function $\nabla_{\boldsymbol{x}} \log q_0(\boldsymbol{x})$. Of course, the objective function in equation 1 entails (i) an expectation over $q_0$ and (ii) the true score $\nabla_{\boldsymbol{x}} \log q_0(\boldsymbol{x})$, which are both not accessible in practice. The expectation can readily be approximated by an average over the data samples $\mathcal{D}_n$; replacing the score needs more care (Vincent, 2011; Song et al., 2020). We come back to this point later in Section 2.2 by representing a time dependent form of denoising score matching (Vincent, 2011). Once the score function is trained, there are various approaches to generate new samples from the target distribution $q_0$ employing the approximated score. These include deterministic and stochastic samplers (see Li et al. (2024b) for an overview), Langevin dynamics among the most popular one Song & Ermon (2019).

## 2.2 DISCRETE-TIME DIFFUSION PROCESS

Let $\boldsymbol{x} \in \mathbb{R}^d$ be an initial data sample and $\boldsymbol{x}_t \in \mathbb{R}^d$ for a discrete time step $t \in \{1, \dots, T\}$ be the latent variable in the diffusion process. Let $\mathbb{Q}_0$ be the initial data distribution, that is, the distribution belonging to the data's density $q_0$, and let $\mathbb{Q}_t$ be the marginal latent distribution in time $t$ in the forward process. We also use the notation $\mathbb{Q}_{t,t+1}$ as the joint distribution over the time $t$ to $t+1$ and $\mathbb{Q} := \mathbb{Q}_{0,\dots,T}$ as the overall joint distribution over the time $T$. In the forward process, white Gaussian noise is gradually added to the data with $\boldsymbol{x}_t = \sqrt{1 - \beta_t} \boldsymbol{x}_{t-1} + \sqrt{\beta_t} \boldsymbol{w}_t$, where $\boldsymbol{w}_t \sim \mathcal{N}(\boldsymbol{0}_d, \mathbb{I}_d)$ and $\beta_t \in (0, 1)$ captures the "amount of noise" that is injected at time step $t$ and are called the noise schedule. This can be written as the conditional distribution

$$\mathbb{Q}_{t|t-1}(\boldsymbol{x}_t | \boldsymbol{x}_{t-1}) = \mathcal{N}\big(\boldsymbol{x}_t; \sqrt{1 - \beta_t} \boldsymbol{x}_{t-1}, \beta_t \mathbb{I}_d\big) .$$

An immediate result is that

$$\mathbb{Q}_t(\boldsymbol{x}_t | \boldsymbol{x}) = \mathcal{N}\big(\boldsymbol{x}_t; \sqrt{\bar{\alpha}_t} \boldsymbol{x}, (1 - \bar{\alpha}_t) \mathbb{I}_d\big) ,$$

for $\alpha_t = 1 - \beta_t$ and $\bar{\alpha}_t = \prod_{i=1}^t \alpha_i$. For large enough $T$ we have $\mathbb{Q}_T \approx \mathcal{N}(\boldsymbol{0}_d, \mathbb{I}_d)$. We also denote $q_t(\boldsymbol{x}_t | \boldsymbol{x})$ as the corresponding density of $\mathbb{Q}_t(\boldsymbol{x}_t | \boldsymbol{x})$ and that $q_t(\boldsymbol{x}_t) = \int q_t(\boldsymbol{x}_t | \boldsymbol{x}) q_0(\boldsymbol{x}) d\boldsymbol{x}$, in which, $q_0$ is the unknown target density for $\boldsymbol{x}$. We also assume that $\mathbb{Q}_0$ is absolutely continuous w.r.t. the Lebesgue measure and so the absolute continuity is preserved for all $t \in \{1, \dots, T\}$ due to the Gaussian nature of the noise. The goal of the reverse process in diffusion models is then to generate samples (approximately) from the distribution $\mathbb{Q}_0$ starting from the Gaussian distribution $\boldsymbol{x}_T \sim \mathcal{N}(\boldsymbol{0}_d, \mathbb{I}_d) =: P_T$. Let's first define

$$\boldsymbol{u}_t(\boldsymbol{x}_t) := \frac{1}{\sqrt{\alpha_t}} \big( \boldsymbol{x}_t + (1 - \alpha_t) \nabla_{\boldsymbol{x}_t} \log q_t(\boldsymbol{x}_t) \big) .$$

At each time step, we then consider the reverse process (for sampling), specifically Langevin dynamics, which can generate samples from a probability density using the true score function, as follows:

$$\boldsymbol{x}_{t-1} = \frac{1}{\sqrt{\alpha_t}} \big( \boldsymbol{x}_t + (1 - \alpha_t) \nabla_{\boldsymbol{x}_t} \log q_t(\boldsymbol{x}_t) \big) + \sqrt{\frac{1 - \alpha_t}{\alpha_t}} \boldsymbol{z}_t = \boldsymbol{u}_t(\boldsymbol{x}_t) + \sigma_t \boldsymbol{z}_t ,$$

for $\boldsymbol{z}_t \sim \mathcal{N}(\boldsymbol{0}_d, \mathbb{I}_d)$ and $\sigma_t := \sqrt{1 - \alpha_t/\alpha_t}$. Let $P_t$ be the marginal distribution of $\boldsymbol{x}_t$ in the true reverse process, which is the reverse process by employing the true scores $\nabla_{\boldsymbol{x}_t} \log q_t(\boldsymbol{x}_t)$, and $p_t$ be the corresponding density. Then, the above statement can be written as $P_{t-1|t} = \mathcal{N}(\boldsymbol{x}_{t-1}; \boldsymbol{u}_t(\boldsymbol{x}_t), \sigma_t^2 \mathbb{I}_d)$. But in practice, one does not have access to the true scores $\nabla_{\boldsymbol{x}_t} \log q_t(\boldsymbol{x}_t)$, instead, an estimate of it namely $\boldsymbol{s}_\Theta(\cdot, t)$, which corresponds to a neural network parameterized with a tuple $\Theta \in \mathcal{B}$ (tuple of weight matrices; we consider ReLU feedforward neural networks with $L$ hidden layers and a total of $p$ parameters), implying

$$\hat{\boldsymbol{u}}_t(\boldsymbol{x}_t) := \frac{1}{\sqrt{\alpha_t}} \big( \boldsymbol{x}_t + (1 - \alpha_t) \boldsymbol{s}_\Theta(\boldsymbol{x}_t, t) \big) .$$

Let $\widehat{P}_t$ be the marginal distribution of $\boldsymbol{x}_t$ in the estimated reverse process implying $\widehat{P}_{t-1|t} = \mathcal{N}(\boldsymbol{x}_{t-1}; \hat{\boldsymbol{u}}_t(\boldsymbol{x}_t), \sigma_t^2 \mathbb{I}_d)$. For $\mathbb{Q}_0$ absolutely continuous, we are then interested in measuring the mismatch between $\mathbb{Q}_0$ and $\widehat{P}_0$ through the Kullback–Leibler divergence

$$D_{\mathrm{KL}}(\mathbb{Q}_0 || P_0) := \mathbb{E}_{X \sim \mathbb{Q}_0} \left[ \log \frac{q_0(X)}{p_0(X)} \right] \geq 0 .$$

## 3 REGULARIZING DENOISING SCORE MATCHING

A promising avenue for accelerating sampling in diffusion models is high-dimensional statistics (Lederer, 2022). High-dimensional statistics is a branch of statistics that deals with many variables. A key idea in high-dimensional statistics is the concept of sparsity; broadly speaking, it means that among those many variables, only few are relevant to a problem at hand. There are different sparsity-related approaches in deep learning, such as dropout (Hinton et al., 2012; Molchanov et al., 2017; Labach et al., 2019; Gomez et al., 2019) or explicit regularization (Alvarez & Salzmann, 2016; Feng & Simon, 2017; Hebiri et al., 2025). The latter approach adds prior functions ("penalties", "regularization") to the objective functions of the estimators. These penalties push the estimators toward specific parts of the parameter space that correspond to certain assumptions, for example, sparsity in $\ell_0$-norm and/or $\ell_1$-norm (Lederer, 2022). The benefits of sparsity are well-documented in regression, deep learning, and beyond (Tibshirani, 1996; Eldar & Kutyniok, 2012; Hastie et al., 2015; Neyshabur et al., 2015; Golowich et al., 2018; Schmidt-Hieber, 2020; Hebiri et al., 2025; Mohades & Lederer, 2023; Golestaneh et al., 2025). However, sparsity-inducing prior functions are abundant in statistics and machine learning, they are rarely employed for generative models (Lin et al., 2016). In this paper, we examine the advantages of incorporating regularization into the objective functions of score-based diffusion models. Additionally, we leverage techniques from empirical process theory van de Geer (2000); Vershynin (2018) to analyze regularized objectives and to calibrate the tuning parameter.

### 3.1 $\ell_1$-REGULARIZED DENOISING SCORE MATCHING

Here we propose an $\ell_1$-regularized estimator for diffusion models, inspired by the concept of "scale regularization" in deep learning (Taheri et al., 2021). Consider the parameter space

$$\mathcal{B}_1 := \left\{ \Theta \in \mathbb{R}^p : \|\Theta\|_1 \leq 1, \ \|s_\Theta(x_t, t)\|_1 \ \leq \ 1 \ \ \forall x_t \in \mathbb{R}^d, t \in \{1, \ldots, T\} \right\}, \tag{2}$$

where $s_\Theta(\cdot, \cdot) : (\mathbb{R}^d, \mathbb{N}) \to \mathbb{R}^d$ is modeled as a neural network with two inputs, parameterized by a tuple $\Theta = (W_0, \ldots, W_L)$. Here, $\Theta$ collects all the weight matrices of the network, which has $L$ hidden layers and input and output dimensions in $\mathbb{R}^d$. We consider $s_\Theta(\cdot, \cdot)$ as a time-dependent score-based model approximating $\nabla_{x_t} \log q_t(x_t)$, which is crucial for sample generation in the backward process of diffusion models. The parameter space $\mathcal{B}_1$ corresponds to sparse networks and sparse score functions, meaning that both the network outputs and the parameters are sparse. Motivated by the denoising score matching objective, a scalable alternative to the objective function in equation 1 (Vincent, 2011), we define a regularized denoising score-matching estimator as

$$(\widehat{\Theta}_{\ell_1}, \hat{\kappa}) \in \underset{\substack{\Theta \in \mathcal{B}_1 \\ \kappa \in (0, \infty)}}{\arg\min} \mathbb{E}_{\substack{t \sim \mathcal{U}_{[0,T]} \\ X_t \sim \mathbb{Q}_t}} \|\kappa s_\Theta(X_t, t) - \nabla_{X_t} \log q_t(X_t)\|^2 + r\kappa^2, \tag{3}$$

where $\kappa \in (0, \infty)$ represents the scale of the score function, $r \in [0, \infty)$ is a tuning parameter that balances the penalty between scale and score matching, and $\mathcal{U}_{[0,T]}$ denotes the uniform distribution over $[0, T]$. The fixed constraint $\Theta \in \mathcal{B}_1$ enforces $\ell_1$-norm regularization, while the actual regularization concerns only on the scale $\kappa \in (0, \infty)$. Additionally, we regularize $\kappa^2$ to simplify our proofs. We will further elaborate in Section A.4 on how the objective function in equation 3 can be computed in practice in terms of expectation and score functions.

Our main contribution in this paper is to theoretically and numerically demonstrate that our proposed regularized estimator in equation 3 can accelerate the sampling process of diffusion models, specifically increasing the rate of convergence in Kullback–Leibler divergence from $d^2/\tau$ (Li et al., 2024b) or $d^{1.5}/\tau$ (Liang et al., 2024b) to $s^2/\tau$, where $s \ll d$.

We are now ready to introduce some assumptions and present our main theorem for our proposed estimator in equation 3.

We first set the learning rates to be used for our theory and analyses. For sufficiently large $T$, we set the step size $\alpha_t$ as

$$1 - \alpha_t \leq c \frac{\log T}{T}, \ \ \forall t \in \{1, \ldots, T\}, \tag{4}$$

for a universal constant $c \in (0, \infty)$, which we omit in the remainder of the paper to simplify notation. We then impose some standard assumptions on the true density function.

**Assumption 1** (Finite second moment). *There exists a constant $M < \infty$ such that $\mathbb{E}_{X_0 \sim \mathbb{Q}_0} \|X_0\|^2 \leq M$.*

Assumption 1 simply states that the distribution is not excessively heavy-tailed; it is applied in the proof of our main theorem. The assumption is standard; see Chen et al. (2023a), Chen et al. (2023d), Benton et al. (2024), Liang et al. (2024b), and many others.

**Assumption 2** (Absolute continuity). *We assume that $\mathbb{Q}_0$ is absolutely continuous w.r.t. the Lebesgue measure, and thus $q_0$ exists.*

We then assume that the derivatives of true log densities are regular, that is, they are bounded by a constant $B \in (0, \infty)$.

**Assumption 3** (Regular derivatives). *For all $t \in \{1, \ldots, T\}$ and $l \in \{1, 2, \ldots\}$ and $\boldsymbol{a} \in [d]^p$ such that $|\boldsymbol{a}|_1 = p \in \{1, 2, \ldots\}$, it holds that*

$$\mathbb{E}_{X_t \sim \mathbb{Q}_t} |\partial_{\boldsymbol{a}}^p \log q_t(X_t)|^\ell \leq B \quad and \quad \mathbb{E}_{X_t \sim \mathbb{Q}_t} |\partial_{\boldsymbol{a}}^p \log q_t(\boldsymbol{u}_t(X_t))|^\ell \leq B,$$

*for a constant $B \in (0, \infty)$.*

The regularity Assumption 3 is required for our analysis in Lemma 1 and also is utilized in previous works like Huang et al. (2024). As discussed extensively in Liang et al. (2024b, Section 5), this assumption is relatively mild, for example for distributions with finite variance or Gaussian mixtures. We then push an assumption over the true gradient vectors $\nabla_{\boldsymbol{x}_t} \log q_t(\boldsymbol{x}_t)$ for $t \in \{1, \ldots, T\}$, that is, assuming they can be well approximated by some sparse versions. More precisely, we assume that only a small subset of features or directions in the high-dimensional space contributes significantly to the score functions. Assumption 4 is well-motivated in high-dimensional statistics and is central to our main Theorem 1.

**Assumption 4** (Sparsity). *There is a sparsity level $s \in \{1, 2, \ldots\}$ and an accuracy $\epsilon \in (0, \infty)$, $\epsilon \leq 1/T$, such that for all $t \in \{1, \ldots, T\}$, there is an analytic auxiliary function $q_t^s(\boldsymbol{x}_t)$ and the corresponding score $\nabla_{\boldsymbol{x}_t} \log q_t^s(\boldsymbol{x}_t)$ that is $s$-sparse and $\epsilon$-accurate:*

$$\mathbb{E}_{X_t \sim \mathbb{Q}_t} \|\nabla_{X_t} \log q_t^s(X_t)\|_0 \leq s \quad and$$

$$\frac{1}{T} \sum_{t=1}^T \sqrt{\mathbb{E}_{X_t \sim \mathbb{Q}_t} \|\nabla_{\boldsymbol{x}_t} \log q_t(X_t) - \nabla_{\boldsymbol{x}_t} \log q_t^s(X_t)\|^2} \leq \epsilon.$$

Our sparsity assumption above is formulated as an *average* over all time steps $t$. It states that, among the many features of the space, only a small subset has a major impact on the score vectors on average, while the majority contribute only marginally ($s < d$). Naturally, the sparsity level $s$ may vary depending on the dataset and could grow at different rates in different contexts. We refer the reader to the detailed discussion following Theorem 1 and in Section A.3, where we analyze the worst-case scenario for the scaling behavior of $\epsilon$ relative to the rate of the tuning parameter $r$ and some simple examples that sparsity holds in practice. Also, we refer to Assumption 5 in the supplementary material for a more relaxed version of our sparsity assumption, which can be used in place of Assumption 4.

**Theorem 1** (Non-asymptotic rates of convergence for regularized diffusion models). *Under the Assumptions 1, 2, 3, and 4 and for $r \geq r^* := C_{\boldsymbol{x}} \sqrt{\log(np)/n}$, our (in-sample) estimator proposed in equation 3 generates samples with*

$$D_{\mathrm{KL}}(\mathbb{Q}_0 \| \widehat{P}_0) \leq \frac{M}{T^2} + \frac{1}{T} \max\{1, 9(sB)^2\} + C_{\boldsymbol{x}} s^2 B^2 \sqrt{\frac{\log(nTp)}{n}} + \Delta_T(\log q, \log q^s)$$

$$+ \inf_{\substack{\Theta \in \mathcal{B}_1 \\ \kappa \in (0, \infty)}} \left\{ \frac{\log T}{T} \sum_{t=1}^T \frac{1}{n} \sum_{i=1}^n \|\kappa \boldsymbol{s}_\Theta(\boldsymbol{x}_t^i, t) - \nabla_{\boldsymbol{x}_t} \log q_t(\boldsymbol{x}_t^i)\|^2 + r\kappa^2 \right\},$$

*for $\Delta_T(\log q, \log q^s) := \sum_{t=1}^T (\mathbb{E}_{X_t \sim \mathbb{Q}_t} [\mathbb{E}_{X_{t-1} \sim \mathbb{Q}_{t-1|t}} [\log q_{t-1}(X_{t-1}) - \log q_{t-1}^s(X_{t-1})] - \mathbb{E}_{X_{t-1} \sim P_{t-1|t}^s} [\log q_{t-1}(X_{t-1}) - \log q_{t-1}^s(X_{t-1})]])$, $C_{\boldsymbol{x}}$ a constant depending on the input distribution, and $p$ denotes the total number of network parameters with probability at least $1 - 1/n$.*

**Corollary 1** (Parametric setting). *Assume that $r = r^*$ and that there exists a pair $(\Theta^*, \kappa^*) \in \mathcal{B}_1 \times (0, \infty)$ such that $\kappa^* s_{\Theta^*}(\boldsymbol{x}_t^i, t) = \nabla_{\boldsymbol{x}_t} \log q_t(\boldsymbol{x}_t^i)$ for all $i \in \{1, \ldots, n\}$ and $t \in \{1, \ldots, T\}$. Then, under the assumptions of Theorem 1, our (in sample) estimator proposed in equation 3 generates samples with*

$$D_{\mathrm{KL}}(\mathbb{Q}_0 || \widehat{P}_0) \leq \frac{1}{T} \max\{1, 9(sB)^2\} + C_{\boldsymbol{x}} \sqrt{\frac{\log(nTp)}{n}} \max\{(sB)^2, (\kappa^*)^2\} + \Delta_T(\log q, \log q^s)$$

*with probability at least $1 - 1/n$.*

Our Theorem 1 reveals that if the true gradient vectors $\nabla_{\boldsymbol{x}_t} \log q_t(\boldsymbol{x}_t)$ can be well approximated by $s$-sparse vectors $\nabla_{\boldsymbol{x}_t} \log q_t^s(\boldsymbol{x}_t)$ and for sufficiently large tuning parameter, the rates of convergence of diffusion models scale with $s$, where potentially $s \ll d$. The term $\Delta_T(\log q, \log q^s)$ in the bound of Theorem 1 measures how close the log density $\log q_t(\boldsymbol{x}_t)$ is to the auxiliary log density $\log q_t^s(\boldsymbol{x}_t)$ across the entire sample space and time steps. Note that we use the notation $P_t^s$ for the distribution of the latent steps in the reverse process, utilizing the sparse scores of Assumption 4 (see also Section B.1 for more details). Following the intuition behind score matching, we argue that Assumption 4 also promotes closeness between these log densities. While one might argue that the sparsity assumption for the score functions at large time steps $t$ and in 4 may not always hold, our detailed discussion in Section A.3 demonstrates that, due to the carefully chosen order of the tuning parameter, our estimator performs comparably to standard score matching even in the worst-case scenario—namely, when the score functions exhibit no sparsity on average. However, when some degree of sparsity is present, our method can lead to significant improvements by promoting sparse representations while still keeping the score estimation error small. Put differently, our estimator strikes a balance between the average score estimation error and the level of sparsity. As a result, it not only achieves low estimation error but also identifies and leverages sparsity level (or scale) when it exists. Furthermore, our empirical observations in Sections 6 and A.3 support the practical validity of our assumptions. Following the work of Karras et al. (2024), we also conjecture that the $\ell_1$-regularizer may serve as a useful tool for improving the training dynamics of diffusion models, although a thorough investigation is still needed. Theorem 1 also directly implies a bound on the the total-variation distance between $\mathbb{Q}_0$ and $\widehat{P}_0$ in view of Pinsker's inequality. Detailed proof of Theorem 1 is provided in Appendix B.2. Corollary 1 follows directly from Theorem 1, so we omit its proof.

Li et al. (2024b); Liang et al. (2024b) show that the reverse diffusion process produces a sample with error roughly at the order of $\varsigma$ if $\sum_{t=1}^{T} \mathbb{E}_{X_t \sim \mathbb{Q}_t} \|\boldsymbol{s}(X_t, t) - \nabla_{X_t} \log q_t(X_t)\|^2 / T \leq \varsigma$. But whether such accurate estimators are available in practice remains unclear. For example, Zhang et al. (2024, Theorem 3.5, Corollary 3.7) upper bounds the estimation error of the score (using $n$ training samples) at the order of $n^{-2\beta/(2\beta+d)}$ assuming the true data distribution $q_0$ is 1. $\sigma_0$-sub-Gaussian and 2. in the Sobolev class of density functions with the order of smoothness $\beta \leq 2$, see also Wibisono et al. (2024); Dou et al. (2024). This result highlights that the original score matching method suffers from the curse of dimensionality (note that $\beta \leq 2 \ll d$). Thus, regularization not only accelerate the reverse process but also help in the estimation of the scores directly—compare to Lederer & Oesting (2023). Block et al. (2020) and Gupta et al. (2024) studied the sample complexity of score matching, providing bounds that scale as $O(d^{5/2}(B'^2 p)^L \sqrt{L}/\tau^2)$ and $O(d^2(Lp) \log B'/\tau^3)$, respectively. Here, $d$ denotes the data dimensionality, $p$ the total number of network parameters (each bounded by $B'$), and $L$ the number of hidden layers of the network. In contrast, our result in Corollary 1 (see also Remark 1) shows a sample complexity scaling as $O(\max\{(sB)^4, (\kappa^*)^4\} \log(pn)/\tau^2)$. This demonstrates a substantial improvement for our regularized estimator: the bounds now depend on the effective sparsity $s \ll d$ and grow only logarithmically with the total number of network parameters, rather than polynomially. More precisely, the bound can even decrease exponentially with the number of hidden layers of the network, which we omit here for simplicity; see Taheri et al. (2021). In fact, regularization can reduce the effective complexity of the network space under consideration, thereby improving the sample complexity of score estimation. This is consistent with the results of Zhu et al. (2023), who establish convergence rates of order $O(d \log \mathcal{N}(\cdot, \mathcal{F})/\tau^2)$, where $\mathcal{N}(\cdot, \mathcal{F})$ quantifies the complexity of the network function class.

**Remark 1** (Sample complexity). *Corollary 1 states that once the network space is sufficiently large (that is, there exists a pair $(\Theta^*, \kappa^*) \in \mathcal{B}_1 \times (0, \infty)$ such that $\kappa^* s_{\Theta^*}(\boldsymbol{x}_t^i, t) = \nabla_{\boldsymbol{x}_t} \log q_t(\boldsymbol{x}_t^i)$ for all $i \in \{1, \ldots, n\}$ and $t \in \{1, \ldots, T\}$), the sample complexity of the regularized diffusion model*

*increases by*

$$O\left( C_{\boldsymbol{x}}^2 \frac{\log(nTp)}{\tau^2} \max\left\{ (sB)^4, (\kappa^*)^4 \right\} \right)$$

*in order to achieve*

$$D_{\mathrm{KL}}(\mathbb{Q}_0 \,\|\, \widehat{P}_0) \leq \tau.$$

We highlight that our regularization technique is motivated by two main considerations: First, it allows for a more focused search during sample generation by concentrating on the features that have the greatest influence on the generated samples (see Figure 1). Second, as noted in (Ren et al., 2025, Example 4.1), one of the dominant sources of error in the convergence of diffusion models is the estimation error

$$\mathcal{E}_{\mathrm{est}}(\Theta) = \mathbb{E}_{\boldsymbol{x}_0 \sim q_0} \left[ \int_0^T \mathbb{E}_{\boldsymbol{x}_t \sim q_{t|0}} \left[ \| s_\Theta(\boldsymbol{x}_t, t) - \nabla \log q_{t|0}(\boldsymbol{x}_t | x_0) \|^2 \right] dt \right],$$

which, in the finite-sample regime, can lead to overfitting. Thus, regularization—being a classical and effective remedy for overfitting—naturally plays a crucial role to reduce estimation error.

## 4 TECHNICAL RESULTS

Here we provide some auxiliary results used in the proof of Theorem 1.

**Lemma 1** (Reverse-step error). *Under the Assumptions 3 and 4 we have*

$$\sum_{t=1}^T \mathbb{E}_{X_t, X_{t-1} \sim \mathbb{Q}_{t,t-1}} \left[ \log \frac{q_{t-1|t}(X_{t-1}|X_t)}{p_{t-1|t}^s(X_{t-1}|X_t)} \right] \leq \frac{1}{T} \left( s^2 B^2 + s^2 B^2 + s^2 B^2 \epsilon + sB\epsilon \right)$$
$$+ \Delta_T(\log q, \log q^s).$$

Lemma 1 helps upper bounding the reverse-step error for the backward process of diffusion models and its detailed proof is provided in Appendix B.3.

We then present a lemma that aids in determining the optimal rates for the tuning parameter.

**Lemma 2** (Empirical processes). *Under the Assumption 4 we obtain*

$$\sup_{\Theta \in \mathcal{B}_1} \left| \mathbb{E}_{X_t \sim Q_t} \| \hat{\kappa} \boldsymbol{s}_\Theta(X_t, t) - \nabla \log q_t^s(X_t) \|^2 - \frac{1}{n} \sum_{i=1}^n \| \hat{\kappa} \boldsymbol{s}_\Theta(\boldsymbol{x}_t^i, t) - \nabla \log q_t^s(\boldsymbol{x}_t^i) \|^2 \right|$$
$$\leq C_{\boldsymbol{x}}(\hat{\kappa}^2 + s^2 B^2) \sqrt{\frac{\log(np)}{n}}$$

*with probability at least $1 - 32/n$, where $C_{\boldsymbol{x}}$ is a constant depending on the input distribution, and $p$ denotes the total number of network parameters.*

Lemma 2 is employed in the proof of Theorem 1 to calibrate the tuning parameter. Its detailed proof is provided in Appendix B.4.

## 5 RELATED WORK

The non-asymptotic rates of convergence for diffusion models established very recently (Block et al., 2020; De Bortoli et al., 2021; De Bortoli, 2022; Lee et al., 2022; Chen et al., 2023c;a; Li et al., 2024b; Chen et al., 2023e; Huang et al., 2024) show the large interest in this topic and are an important step forward but do not fully explain the success of generative models either. For example, results like Block et al. (2020, Theorem 13) provide rates of convergence for diffusion models in terms of Wasserstein distance employing the tools from empirical-process theory, but they suffer from the curse of dimensionality in that the rates depend exponentially on the dimensions of the data, that is, the number of input features. Recent works then concentrate on improving convergence guarantees to grow polynomially in the number of input features under different assumptions on the original and estimated scores ($L_2$-accurate score estimates, Lipschitz or smooth scores, scores with bounded

moments) (Lee et al., 2022; Wibisono & Yang, 2022; Chen et al., 2023d;c;a;e; Lee et al., 2023; Huang et al., 2024). For example, Lee et al. (2022) prove a convergence guarantee in terms of total variation for SGMs, which has a polynomial dependence on the number of input features if the score estimate is $L_2$-accurate for any smooth distribution satisfying the log-Sobelev inequality. A very recent work by Li et al. (2024b) proposes improved convergence rates in terms of total variation for DDPMs with ordinary differential equations and stochastic differential equations samplers that are proportional to $d^2/\tau$, where $d$ is the number of input features and $\tau$ the error in the measure under consideration. They assumed 1. finite support assumption, 2. $L_2$-accurate score estimates, and 3. accurate Jacobian matrices. Li et al. (2024b, Theorem 3) also provides rates growing by $d^3/\sqrt{\tau}$ for an accelerated ordinary differential equations samplers.

While works like Li et al. (2024b) and Li et al. (2024a) concentrate more on improving the rates in $\tau$, Chen et al. (2023c) focus on improving the rates in $d$ for denoising diffusion implicit models. Chen et al. (2023c) use a specially chosen corrector step based on the underdamped Langevin diffusion to achieve their improvements, namely rates proportional to $L^2\sqrt{d}/\tau$ by assuming: 1. the score function along the forward process is $L$-Lipschitz, 2. finite second moments of the data distribution, and 3. $L_2$-accurate score estimates. Chen et al. (2023a); Benton et al. (2024) then relaxed the assumptions over the data distribution and proposed the rates of convergence for DDPM proportional to $\sqrt{d^2/\tau}$ and $\sqrt{d/\tau}$ under 1. finite second moments of the data distribution, and 2. $L_2$-accurate score estimates. Further research directions may also build upon Chen et al. (2023b), who assume that the data lie on a low-dimensional linear subspace. They demonstrate that, in this setting, the convergence rates depend on the dimension of the subspace.

## 6    EMPIRICAL SUPPORT

In this section, we demonstrate the benefits of regularization for diffusion empirically. Rather than relying on large-scale pipelines and data, which are subject to a number of other factors, we study the influence of regularization in simple, well-explored setups. We present a toy example and the MNIST family dataset here, deferring additional simulations and setups to Appendix Section A, where we study more complicated datasets including FashionMNIST, Butterflies, and CIFAR10 datasets.

### 6.1    TOY EXAMPLE

We first highlight the influence of regularization on the sampling process of a 3D toy example. We consider 2000 three-dimensional, independent Gaussian samples with mean zero and covariance matrix $[0.08, 0, 0; 0, 1, 0; 0, 0, 1]$; hence, the data fluctuate most around the $y$ and $z$ axes. We then train two diffusion models, with the same data: the original denoising score matching and the same with an additional sparsity-inducing regularization (as proposed in equation 3) and $r = 0.001$. Figure 1 visualizes the data (first panel) and the sampling process with $T = 60$ for original score matching (second panel) and the regularized version (third panel). Both models start from the blue dot. The figure shows that the regularized version provides a more focused sampling.

### 6.2    MNIST

We now compare original score matching and the regularized version on MNIST dataset (LeCun et al., 1998) including $n = 50\,000$ training samples. We are interested to time steps $T \in \{500, 50, 20\}$ for sampling and we consider regularization with $r = 0.0005$ for $T = 500$ and $r = 0.003$ for $T \in \{50, 20\}$. In fact, we set the tuning parameter as a decreasing function of $T$, let say $r = f[T] \in O(c'/T)$ for a real constant $c' \in (0, \infty)$. Note that two models are already trained over the same amount of data and identical settings employing different objectives. For sampling (starting from pure noise), then we try different values of time steps $T \in \{500, 50, 20\}$. That means, for small values of $T$, we just need to pick up a **larger step size** as we always start from pure noise (see Algorithm 2). Figure 2 shows the results. While the original score matching fails to generate reasonable samples for small values of $T$, our proposed score function performs successfully even for $T = 20$. Let note that generating 64 samples with $T = 500$ steps takes about 11 seconds, while using $T = 50$ steps can reduce runtime to under 1 second, a tenfold speedup. In all our simulations, we use the same network structure, optimization method, and sampling approach with identical settings for both approaches (see Appendix A.5 for detailed settings). The only difference lies in the objective functions: one is

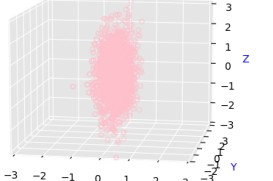 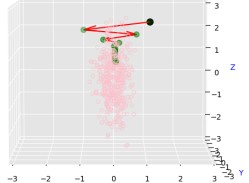 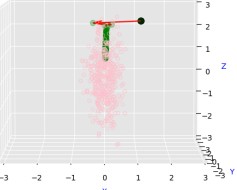

Figure 1: Visualizing the sampling process for 3D data (first panel) with original denoising score matching (second panel) versus regularized denoising score matching (third panel). The original samples are depicted as red circles, blue circles indicate the starting points for sampling, and green circles represent the latent generated samples. The red arrows illustrate the sampling paths. It is evident that regularized denoising score matching predominantly adheres to the two-dimensional sub-manifold (along the $Y$ and $Z$ axes), whereas the original denoising score matching explores the entire 3D space.

regularized, while the other is not. While it could be argued that alternative network structures or sampling processes might enhance the quality of the generated images for original score matching, our focus remains on the core idea of regularization fixing all other factors and structures. We defer the enhanced versions of our simulations aimed at achieving higher-quality images to future works.

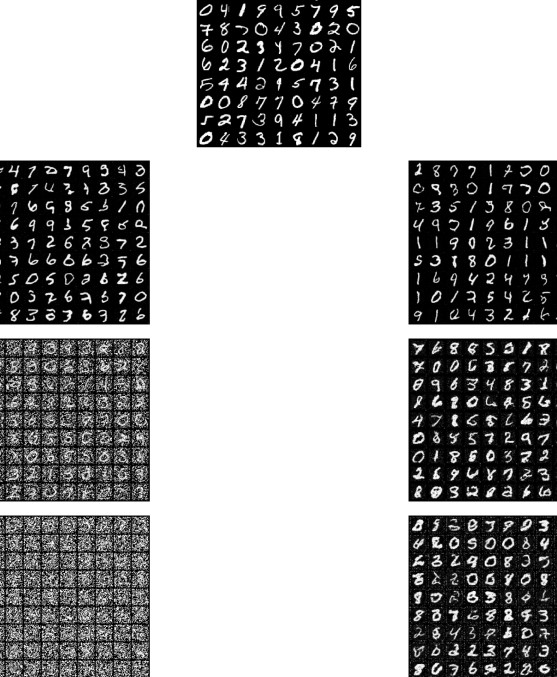

Figure 2: Image generation using the original denoising score matching (left column) versus the regularized version (right column) for different time steps, $T = 500, T = 50$, and $T = 20$ (from top to bottom). The middle column displays 81 original samples from the MNIST dataset for comparison with images of dimensions $d = 28 \times 28 \times 1 = 784$.

## 6.3 FASHIONMNIST

We follow almost all the settings as in Section 6.2 with $r = 0.0001$ for $T = 500$ and $r = 0.002$ for $T \in \{70, 50\}$ for `FashionMNIST` dataset (Xiao et al., 2017) including $n = 50\,000$ training samples. Results are deferred to Figure 3 in the Appendix. Following the generated images obtained using both approaches, and ensuring that all factors except the objective functions remain identical, we observe that the original score-matching approach produces samples that appear oversmoothed and exhibit imbalanced distributions (see the first image of the left panel of Figure 3). In contrast, our regularized approach with a considerably small tuning parameter, generates images that resemble the true data more closely and exhibit a more balanced distribution (see the first image of the right panel of Figure 3). For instance, the percentages of generated images for Sandals, Trousers, Dresses, Ankle Boots, and Bags are approximately $(0.0, 0.7, 2.0, 3.0, 4.0)$ using the original score matching, compared to $(8.0, 6.0, 8.0, 10.0, 10.0)$ with the regularized version, highlighting the clear imbalance in distribution for the original score matching.

## 7 CONCLUSION

Our mathematical proofs (Section 3) and empirical illustrations (Section 6) demonstrate that regularization can reduce the computational complexity of diffusion models considerably. Broadly speaking, regularization replaces the dependence on the input dimension by a dependence on a much smaller intrinsic dimension. But our findings might just be the beginning: we believe that types of regularization beyond the sparsity-inducing $\ell_1$-regularization applied here, such as total variation, could lead to further improvements. Finally, exploring sparsity in more structured domains—such as wavelet or Fourier bases, where many signals are naturally sparse—offers a promising and interpretable direction for future work.

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

## A   APPENDIX 1: ADDITIONAL SIMULATIONS AND TECHNICAL RESULTS

Here, we present additional simulations and technical results. We provide additional simulation supporting our theories on further image dataset Butterflies in Section A.1 and CIFAR10 in Section A.2. We then provide a detailed discussion over Assumption 4 in Section A.3. We introduce our training and sampling approach in Section A.4 and provide details about network architecture and training settings in Section A.5. We also conducted additional simulations to compare the perfomance of our method vs using other sparsity-inducing regularizers in Section A.6.

### A.1   BUTTERFLIES

We also compare original diffusion and regularized analog on `Butterflies` dataset (smithsonian-butterflies) including $n = 10\,000$ training samples. We consider regularization $r = 0.0001$ for $T = 1000$ and $r = 0.0005$ for $T \in \{200, 150\}$. Results are provided in Figure 4. Again, our results show that our approach perform better than the original score matching for small values of $T$.

### A.2   CIFAR10

We compare original diffusion and regularized analog on `CIFAR10` dataset with $n = 50\,000$ training samples. We consider regularization $r = 0.0001$ for $T \in \{1000, 500, 200\}$. Results are provided in Figure 5. As shown in the images, our regularized version with $r = 0.0001$ generates high-quality images for $T = 1000$ and still performs better than original score matching for $T = 500$ and $T = 200$. For our CIFAR10 simulations, we used the original code provided by Song et al. (2021b) as our baseline. However, due to hardware limitations, we reduced the model complexity by using significantly fewer channels (32 instead of 128 as in the original model). We also used the *Fréchet Inception Distance* (FID) as a quantitative measure to evaluate the quality of generated samples for CIFAR10 data. For $T = 1000$, our method achieves $\text{FID}_{\ell_1} = 28$, compared to $\text{FID}_o = 32$ for standard score matching. For $T = 500$, we obtain $\text{FID}_{\ell_1} = 65$, while the original score matching yields $\text{FID}_o = 78$. These results demonstrate that our approach consistently outperforms standard score matching, even with relatively large time steps $T$.

We have now repeated the experiments using a larger model with a base channel size of 128. For clarity of comparison, we report a collection of generated car images in Figure 6. Moreover, we computed the FID scores for both models. The original score-matching setup achieves an FID of 25,

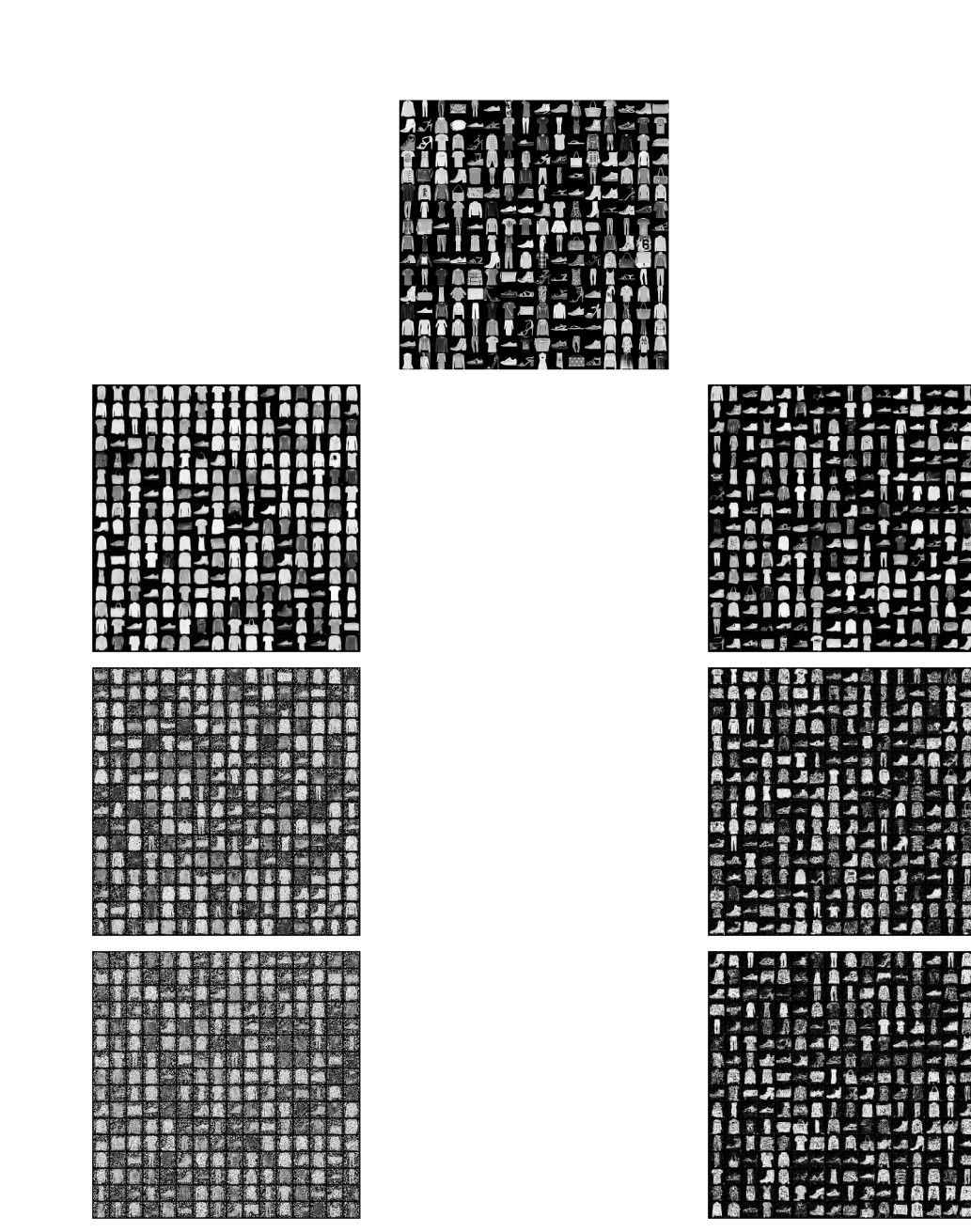

Figure 3: Image generation using the original denoising score matching (left column) versus the regularized version (right column) for different time steps, $T = 500, T = 70$, and $T = 50$ (from top to bottom). The middle column displays 256 original samples from the `FashionMNIST` dataset for comparison with images of dimensions $d = 28 \times 28 \times 1 = 784$. Our regularized version generates high-quality images for $T = 500$ (comparable to the original denoising score matching) and still produces good images even for samll $T$, while the original denoising score matching totally fails. Another notable observation is that our regularization results in more balanced image generation, as evident when comparing our method to the original denoising score matching at $T = 500$, where the latter produces overly smooth images.

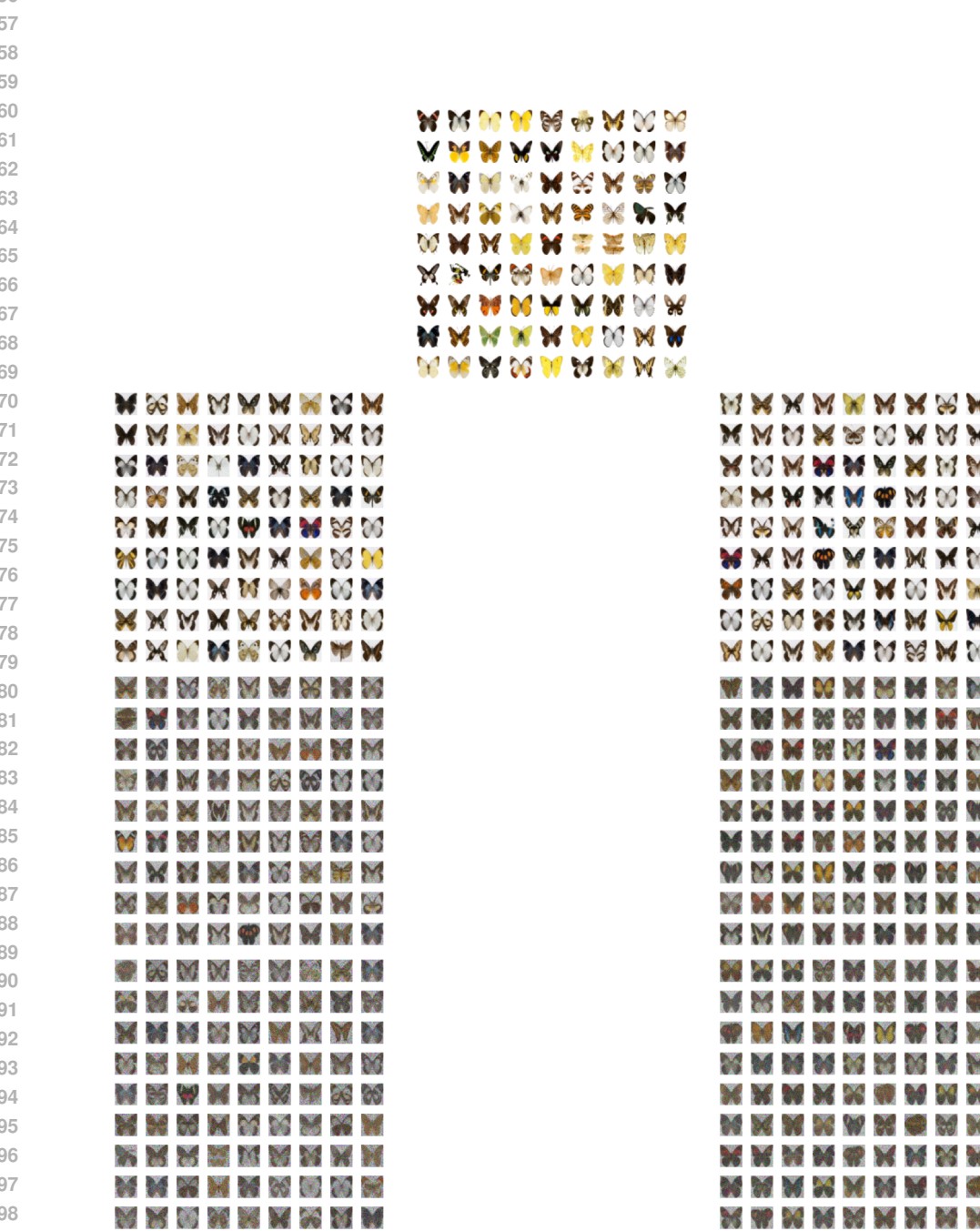

Figure 4: Image generation using the original denoising score matching (left column) versus the regularized version (right column) for different time steps, $T = 1000, T = 200$, and $T = 150$ (from top to bottom). The middle column displays 81 original samples from the Butterflies dataset for comparison. The dataset consists of images with dimensions $d = 28 \times 28 \times 3 = 2352$. As shown in the images, our regularized version generates high-quality images for $T = 1000$ (comparable to the original denoising score matching) and still perform better than original denoising score matching for $T = 200$ and $T = 150$.

consistent with the results reported in Song & Ermon (2019). While we were able to further improve the FID to 23 employing our regularized objective with $r = 1e - 4$. For $T = 500$, our regularized approach ends with FID $= 25$, while original score matching drops to 38. For small time budget, $T = 200$, the original score matching drops in FID to 160, while our approach ends with FID $= 49$ for $r = 5e - 4$.

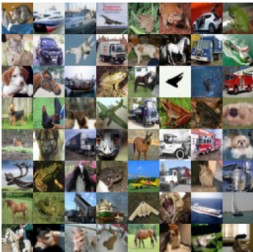

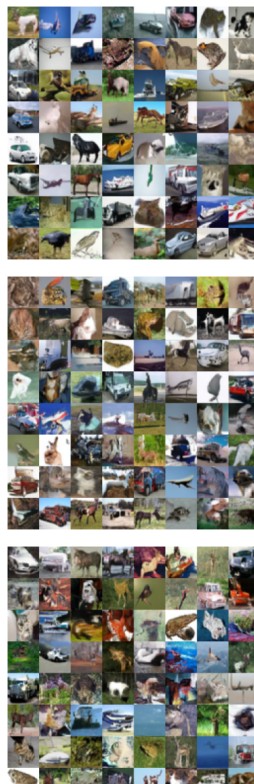
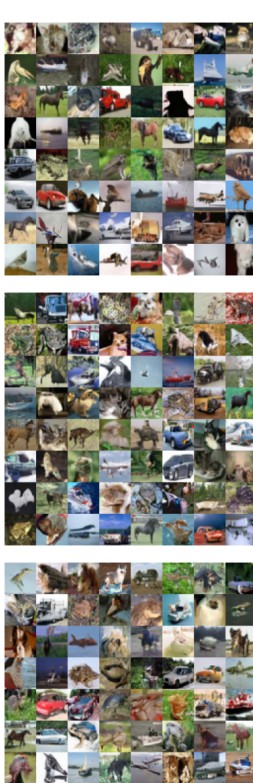

Figure 5: Image generation using the original denoising score matching (left column) versus the regularized version (right column) for different time steps, $T = 1000, T = 500$, and $T = 200$ (from top to bottom). The middle column displays 64 original samples from the CIFAR10 dataset for comparison. The dataset consists of images with dimensions $d = 32 \times 32 \times 3 = 3072$. As shown in the images, our regularized version with $r = 0.0001$ generates high-quality images for $T = 1000$ and still performs better than original score matching for $T = 500$ and $T = 200$.

## A.3 DISCUSSION OVER ASSUMPTION 4

We consider a worst-case scenario for our estimator and examine the validity of Assumption 4. According to the statement of Theorem 1, let set the tuning parameter as $r = r^* = \max(1/T, \sqrt{\log(np)/n})$, where we omit constants for simplicity here. Then, the objective function seeks a pair $(\hat{\kappa}, \hat{\Theta})$ such

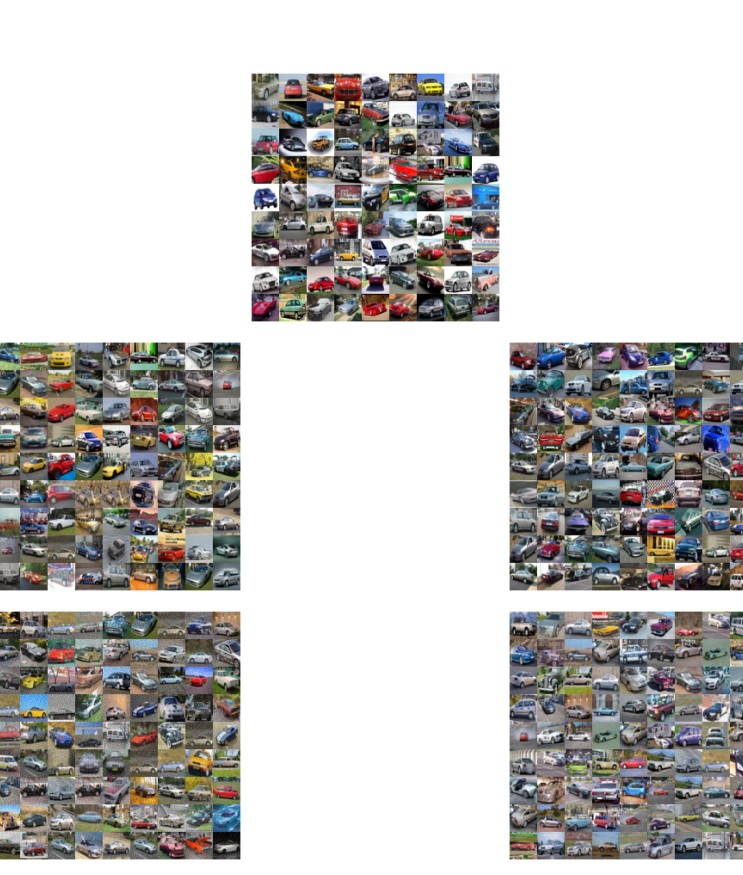

Figure 6: Image generation using the original denoising score matching (left column) versus the regularized version (right column) for different time steps, $T = 100$ and $T = 200$ (from top to bottom). The middle column displays 81 original samples from the `CIFAR10` dataset for comparison. The dataset consists of images with dimensions $d = 32 \times 32 \times 3 = 3072$. As shown in the images, our regularized version with $r = 0.0001$ generates high-quality images for $T = 1000$ and still performs better than original score matching for $T = 200$.

that

$$(\hat{\kappa}, \hat{\Theta}) \in \arg\min_{\kappa, \Theta \in \mathcal{B}_1} \frac{1}{T} \sum_{t=1}^{T} \left( \frac{1}{n} \sum_{i=1}^{n} \left| \kappa s_\Theta(x_t^i, t) - \nabla \log q_t(x_t^i) \right|^2 \right) + r^* \kappa^2.$$

Suppose $(\kappa_o, \Theta_o)$ is the estimator from the original score matching, i.e., it perfectly fits the true score function. Then,

$$\frac{1}{T} \sum_{t=1}^{T} \left( \frac{1}{n} \sum_{i=1}^{n} \left| \kappa_o s_{\Theta_o}(x_t^i, t) - \nabla \log q_t(x_t^i) \right|^2 \right) + r^* \kappa_o^2 \approx 0.0 + r^* \kappa_o^2.$$

While our estimator may still yield a smaller objective value compared to the original score matching implying

$$\inf_{\kappa, \Theta \in \mathcal{B}_1} \frac{1}{T} \sum_{t=1}^{T} \left( \frac{1}{n} \sum_{i=1}^{n} \left| \kappa s_\theta(x_t^i, t) - \nabla \log q_t(x_t^i) \right|^2 \right) + r^* \kappa^2 \le r^* \kappa_o^2.$$

Therefore, for our estimator $(\widehat{\Theta}_{\ell_1}, \hat{\kappa})$ we have

$$\frac{1}{T} \sum_{t=1}^{T} \left( \frac{1}{n} \sum_{i=1}^{n} \left| \hat{\kappa} s_{\widehat{\Theta}_{\ell_1}}(x_t^i, t) - \nabla \log q_t(x_t^i) \right|^2 \right) + r^* \hat{\kappa}^2 \le r^* \kappa_o^2$$

that implies $\hat{\kappa} < \kappa_o$. Note that the scale term reflects the sparsity level since it determines the radius of the optimal unit ball. Despite this reduced scale, the estimator still achieves small score estimation error:

$$\frac{1}{T} \sum_{t=1}^{T} \left( \frac{1}{n} \sum_{i=1}^{n} \left| \hat{\kappa} s_{\widehat{\Theta}_{\ell_1}}(x_t^i, t) - \nabla \log q_t(x_t^i) \right|^2 \right) < r^* \kappa_o^2 \le \max\left( \frac{1}{T}, \frac{\sqrt{\log(np)}}{\sqrt{n}} \right) \kappa_o^2.$$

These last two points show that our regularized estimator either performs comparably to the original score matching estimator (if no better $(\hat{\kappa}, \widehat{\Theta}_{\ell_1})$ exists), or it finds a better pair with smaller scale and only a small estimation error for score functions. In such cases, we may interpret $\hat{\kappa} s_{\widehat{\Theta}_{\ell_1}}(\boldsymbol{x}_t^i, t)$ as a sparse estimator for $\nabla \log q_t$ in Assumption 4, since $\hat{\kappa} < \kappa_o$, indicating a smaller radius for the unit ball, which gives a small estimation error for scores. Although Assumption 4 is stated for the expected error, not the in-sample one, this still gives a meaningful interpretation in practice. We would also like to highlight the expressive power of the network space $\kappa s_\Theta(\cdot)$ with $\Theta \in \mathcal{B}_1$, which, following the original work by Taheri et al. (2021), can approximate the entire network space $s_\Omega$ under some assumptions, where $\Omega$ denotes an unconstrained parameter space. This is related to how interpret $\hat{\kappa} s_{\widehat{\Theta}_{\ell_1}}(.)$ as $\nabla \log q^s(.)$. Our discussion above shows that our estimator may behave, in the worst case, similarly to the original score matching. However, it can perform significantly better in the sense that it induces sparsity (by finding a smaller scale $\hat{\kappa} \le \kappa_o$) while still achieving a small estimation error for the score of the order of $\max(1/T, 1/\sqrt{n})\hat{\kappa}^2$ (note that $\hat{\kappa} \approx sB$).

Also, as an extreme and simple case of sparse scores one can consider a two-dimensional dataset, where the distribution along the $x$-axis is Gaussian, while along the $y$-axis it is uniform and $x$ and $y$ are independent. The score function is sparse, given by $\nabla \log p(x, y) = (\partial_x \log p(x), 0)$, indicating no contribution from the $y$-direction. This toy example can be naturally extended to more complex datasets, such as images, where includes local regularity like certain regions (e.g., background pixels) exhibit near-uniform distributions and thus contribute negligibly to the score. Such structure leads to sparsity in the score function, which is consistent with our assumption, and in fact our empirical observations already well support that.

**Assumption 4 on toy data**    We conducted an experiment to train a diffusion model for generating mixed Gaussian–Uniform data in a $d$-dimensional space. We varied the number of Gaussian features $s$, with the remaining $d - s$ dimensions drawn from a uniform distribution independent of the other features. For each setting, we trained a diffusion model and evaluated the quality of the generated samples by computing the KL divergence between the generated and original distributions. Results show that our regularized approach, using a fixed tuning parameter $\lambda = 0.001$, not only outperforms standard score matching when the sparsity level is high (i.e., many features are uniform), but also consistently dominates standard score matching even when the sparsity level is low.

**Assumption 4 on MNIST**  We conducted an experiment on MNIST to validate the sparsity assumption for a diffusion model trained with standard score matching. We measured score approximation error across all time steps for sparsity levels $\{600, 400, 200, 100, 20\}$, obtaining overall errors of approximately $\{0.05, 0.09, 0.18, 0.30, 1.0\}$. Errors were also computed separately for early and late time steps: early steps gave $\{0.01, 0.05, 0.10, 0.17, 0.61\}$, while late steps gave $\{0.32, 0.46, 0.89, 2.9, 5.1\}$. Since early steps dominate in diffusion models, these results indicate that the sparsity assumption holds well in practice, with negligible bias up to moderate sparsity levels. Furthermore, MNIST image generation with small $T$ shows even smaller bias than suggested by the above errors, likely due to score estimation and optimization effects, and our method performs well even for high sparsity.

We would note that Assumption 4 can also be relaxed as:

**Assumption 5** (Relaxed-sparsity). *There is a sparsity level $s \in \{1, 2, \dots\}$ and an accuracy $\epsilon \in (0, \infty)$, $\epsilon \leq 1/s^a T^{a'}$ ($a', a \in (0, \infty)$), such that for all $t \in \{1, \dots, T\}$, there is an analytic auxiliary function $q_t^s(\boldsymbol{x}_t)$ and the corresponding score $\nabla_{\boldsymbol{x}_t} \log q_t^s(\boldsymbol{x}_t)$ that is $s$-sparse and $\epsilon$-accurate:*

$$\mathbb{E}_{X_t \sim \mathbb{Q}_t} \|\nabla_{X_t} \log q_t^s(X_t)\|_0 \ \leq \ s \ \ and$$

$$\frac{1}{T} \sum_{t=1}^{T} \sqrt{\mathbb{E}_{X_t \sim \mathbb{Q}_t} \|\nabla_{\boldsymbol{x}_t} \log q_t(X_t) - \nabla_{\boldsymbol{x}_t} \log q_t^s(X_t)\|^2} \ \leq \ \epsilon \, .$$

Collecting the arguments above, we conclude that by assuming sparsity in the score functions, we incur a bias term of order $1/s^a T^{a'}$; however, this trade-off allows us to improve the overall convergence rates to $O(s/\sqrt{n} + 1/s^a T^{a'})$, in contrast to the $O(d/\sqrt{n})$ rates established in Zhu et al. (2023). By selecting the tuning parameter appropriately, we ensure that the trade-off between bias and variance is well balanced.

Recently, regularization has been employed to prevent memorization (Gabriel et al., 2025; Baptista et al., 2025). The analysis in Baptista et al. (2025) highlights the necessity of regularization to avoid reproducing the analytically tractable minimizer, and in doing so, lays the groundwork for a principled understanding of how to regularize effectively. Their numerical experiments investigate the properties of: (i) Tikhonov regularization; (ii) regularization schemes designed to promote asymptotic consistency; and (iii) implicit regularizations induced either by under-parameterization of a neural network or by early stopping during training.

**Sparsity within scores**  To quantitatively assess sparsity in the score fields, we compute for each MNIST and CIFAR10 images the fraction of score coefficients whose absolute value falls below a small threshold $(0.01)$, for every time step $t$ and for both the regularized and unregularized models. Concretely, we flatten each score tensor into a vector and measure, per image, the proportion of near-zero entries (per model per time step). The resulting distributions are summarized using box plots, which visualize—across the entire dataset—how much of the score field is effectively sparse and allow for a direct comparison between the baseline model and our sparsity-regularized variant (see Figure 7). We also obtain score heat maps by computing the pointwise norm of the estimated score field for each pixel for CIFAR10 and MNIST in Figure 8. This highlights regions where the model assigns high sensitivity, revealing structural or edge-related features. Our simulations reveal three key observations: (1) both models exhibit a fraction of near-zero score coefficients, especially at small time steps where the perturbed image retains meaningful structure; (2) the regularized model consistently produces a much larger proportion of near-zero entries, demonstrating significantly stronger sparsity; and (3) this behavior is fully consistent with our earlier discussion of local regularity—such as background regions and near-uniform pixel areas—which naturally contribute very little to the score field; see Figure 8, where background pixels exhibit highly sparse scores while structurally important edge pixels remain distinctly non-zero in early time steps. We also remark that the observed sparsity depends on the chosen threshold value. In our case, we selected a very small threshold to demonstrate that even with such a small cutoff, a level of sparsity still appears, while larger thresholds yield an even higher degree of sparsity. Let us also remark that, as discussed earlier, when there is no considerable sparsity in the data, our method performs comparably to the original score matching. This is clearly reflected in the box plots at later time steps (Figure 7).

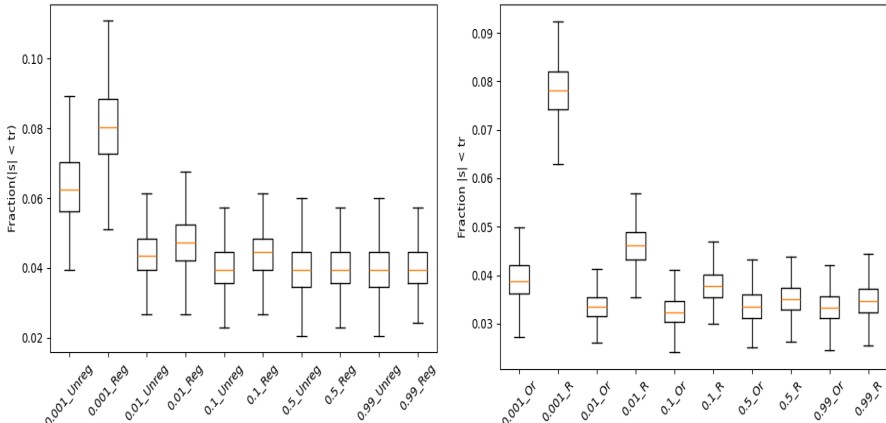

Figure 7: Box plot summarizing the distribution of small score magnitudes for original and regularized model over time for MNIST (left panel) and CIFAR10 (right panel).

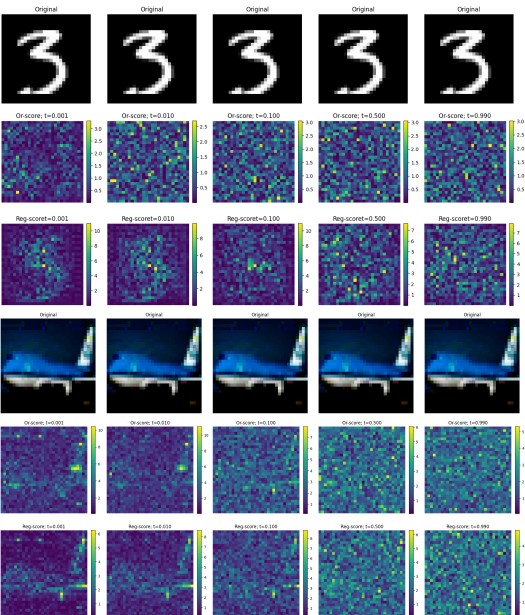

Figure 8: Heat map of pixel-wise score magnitudes, highlighting informative regions where the model assigns high sensitivity.

### A.4 TRAINING AND SAMPLING ALGORITHMS

Here we provide details about how we solve the objective function equation 3 in practice, that is, how we deal with the expected values and score functions.

Let first define the objective function over a batch of training examples $\boldsymbol{x}_{b_{\mathrm{s}}}$ (a batch of size $b_{\mathrm{s}} \in \{1, 2, \dots\}$) and for a batch of random time steps $\boldsymbol{t}_{b_{\mathrm{s}}} \in (0, 1]^{b_{\mathrm{s}}}$:

$$f(\kappa, \Theta, \boldsymbol{x}_{b_{\mathrm{s}}}, \boldsymbol{t}_{b_{\mathrm{s}}}) := \frac{1}{b_{\mathrm{s}}} \sum_{i=1}^{b_{\mathrm{s}}} \|\kappa \boldsymbol{s}_{\Theta}(\boldsymbol{x}_{t_i}^i, t_i) - \nabla_{\boldsymbol{x}_t} \log q_t(\boldsymbol{x}_{t_i}^i | \boldsymbol{x}^i)\|^2] + r\kappa^2 \tag{5}$$

with

$$\mathbb{Q}_t(\boldsymbol{x}_{t_i}^i | \boldsymbol{x}^i) = \mathcal{N}(\boldsymbol{x}^i, \sigma_{t_i} \mathbb{I}_d) \tag{6}$$

with $\sigma_t := (\sigma^{2t} - 1)/(2 \log \sigma)$ for $t \in (0, 1]$ and a large enough $\sigma \in (0, \infty)$ (we set $\sigma = 5$ for `Butterflies` and `CIFAR10` and $\sigma = 25$ for other datasets). Here $\boldsymbol{x}_t^i$ corresponds to a perturbed

version of the training sample $\boldsymbol{x}^i$ ($i$th sample of the batch) in time step $t$. As stated in equation 6, once $\sigma$ is large, $\boldsymbol{x}_1$ ($t = 1$) goes to a mean-zero Gaussian. And as shown in Vincent (2011), the optimization objective $\mathbb{E}_{q_t(\boldsymbol{x}_t|\boldsymbol{x})q_0(\boldsymbol{x})}[\|\kappa\boldsymbol{s}_\Theta(\boldsymbol{x}_t, t) - \nabla_{\boldsymbol{x}_t}\log q_t(\boldsymbol{x}_t|\boldsymbol{x})\|^2]$ for a fixed variance $\sigma_t$ is equivalent to the optimization objective $\mathbb{E}_{q_t(\boldsymbol{x}_t)}[\|\kappa\boldsymbol{s}_\Theta(\boldsymbol{x}_t, t) - \nabla_{\boldsymbol{x}_t}\log q_t(\boldsymbol{x}_t)\|^2]$ and, therefore, satisfies $\kappa^*\boldsymbol{s}_{\Theta^*}(\boldsymbol{x}_t, t) = \nabla_{\boldsymbol{x}_t}\log q_t(\boldsymbol{x}_t)$. We then provide Algorithm 1 for solving the objective function in equation 3. Note that we can easily compute the score functions in equation 5 since there is a closed-form solution for them as densities are just Gaussian conditional on $\boldsymbol{x}^i$.

---

**Algorithm 1** Training algorithm

1: **Inputs:** $\sigma$, $n_{\text{epochs}}$ (number of epochs), $b_{\text{s}}$(batch-size), eps $= 0.00001$
2: **Outputs:** $(\widehat{\Theta}_{\ell_1}, \hat{\kappa})$
3: Initialize parameters $(\widehat{\Theta}_{\ell_1}, \hat{\kappa})$
4: **for** $i = 1$ to $n_{\text{epochs}}$ **do**
5:     **for** $\boldsymbol{x}_{b_{\text{s}}}$ in data-loader **do**
6:         $\boldsymbol{t}_{b_{\text{s}}} = \{\mathcal{U}_{[0,1]}\}^{b_{\text{s}}}(1 - \text{eps}) + \text{eps}$
7:         One step optimization minimizing $f(\kappa, \Theta, \boldsymbol{x}_{b_{\text{s}}}, \boldsymbol{t}_{b_{\text{s}}})$ in equation 5 employing a random batch
        of time steps $\boldsymbol{t}_{bs} \in (0, 1]^{bs}$ and updating $(\widehat{\Theta}_{\ell_1}, \hat{\kappa})$
8:     **end for**
9: **end for**

---

Parameter eps in Algorithm 1 is introduced for numerical stability and to refuse $t = 0$. For a sufficiently large number of epochs, we expect to learn the scores accurately for different time steps. For sampling process, we employ a naive sampler as proposed in Algorithm 2 employing Langevin dynamics (Song & Ermon, 2019, Section 2.2) to align with our theory.

---

**Algorithm 2** Sampling algorithm

1: **Inputs:** $\sigma$, eps $= 0.00001$, T (Time steps)
2: **Output:** $\boldsymbol{x}$
3: $\boldsymbol{x} = \boldsymbol{x}_{\text{init}} = \mathcal{N}(\boldsymbol{0}_d, \mathbb{I}_d)\sigma_1$
4: $\boldsymbol{t} = linspace(1., eps, T)$ (make a grid of time steps)
5: $\eta = \boldsymbol{t}[0] - \boldsymbol{t}[1]$ (set step size)
6: **for** t in $\boldsymbol{t}$ **do**
7:     $\boldsymbol{x} = \boldsymbol{x} + \eta\hat{\kappa}\boldsymbol{s}_{\widehat{\Theta}_{\ell_1}}(\boldsymbol{x}, t) + \sqrt{2\eta}\mathcal{N}(\boldsymbol{0}_d, \mathbb{I}_d)$ (update $\boldsymbol{x}$)
8: **end for**

---

A.5   Network architecture and training settings

Our model is a U-Net architecture with 4 downsampling and 4 upsampling layers, each comprising residual blocks. The network starts with a base width of 32 channels, doubling at each downsampling step to a maximum of 256 channels, and mirrors this in the decoder. A bottleneck layer with 256 channels connects the encoder and decoder. Time information is encoded using Gaussian Fourier projections and injected into each residual block via dense layers. Group normalization is applied within the residual blocks, and channel attention mechanisms are included selectively to enhance feature representations. For training, we used the Adam optimizer with a learning rate of 0.001, and for sampling, we employed a signal-to-noise ratio of 0.1. We used a batch size of 128 and trained for 2000 epochs on the `Butterflies` dataset and less than 1000 epochs on the other datasets.

A.6   Comparison with other sparsity-inducing regularizers

We conducted additional simulations to compare the perfomance of our method vs using other sparsity-inducing regularizers.

    1. $\ell_2$-**regularization:** Although the $\ell_2$-norm helps control the growth of magnitudes, it does not promote sparsity by design. Our complementary simulations show that using $\ell_2$-regularization does not significantly speed up inference time in our experiments.

2. **Group lasso:** Our simulations show that applying group lasso on images—considering groups as $p \times p$ pixel blocks ($4 \times 4$)—can also improve inference time and, for some datasets like MNIST, even outperform $\ell_1$-regularization.

3. **Combination of norms:** Employing a combination of norms, namely group lasso plus $\ell_1$-norm, performs comparably to individual regularizations for the MNIST family, but we do not observe any considerable improvement over using either group lasso or $\ell_1$-norm alone.

## A.7 RUNTIMES

We measured the optimization time for training the regularized and the original diffusion models on MNIST over 50 epochs using Adam on the same device. The training time for the original DFM is approximately 20 minutes in total, while the regularized version takes about 21 minutes. Both objectives converge after about 20 epochs. This shows that adding regularization has only a minor effect on the training time. Similar behavior also observed for other datasets.

## B APPENDIX 2: AUXILIARY RESULTS AND PROOFS

In this section, we first provide two auxiliary results and then, we present detailed proofs of our main results. Note that throughout our proofs, we will omit the index $\boldsymbol{x}_t$ from $\nabla_{\boldsymbol{x}_t} \log q_t(\cdot)$ for simplicity in notations and simply write $\nabla \log q_t(\cdot)$.

### B.1 AUXILIARY RESULTS

We denote the distribution of the latent steps in the reverse process, utilizing the sparse gradient vector, as $P^s_{t-1|t} := \mathcal{N}(\boldsymbol{x}_{t-1}; \boldsymbol{u}^s_t(\boldsymbol{x}_t), \sigma^2_t \mathbb{I}_d)$ with

$$\boldsymbol{u}^s_t(\boldsymbol{x}_t) := \frac{1}{\sqrt{\alpha_t}}\big(\boldsymbol{x}_t + (1 - \alpha_t)\nabla_{\boldsymbol{x}_t} \log q^s_t(\boldsymbol{x}_t)\big). \tag{7}$$

We then connect the conditional distributions $q_{t-1|t}(\boldsymbol{x}_{t-1}|\boldsymbol{x}_t)$ and $p^s_{t-1|t}(\boldsymbol{x}_{t-1}|\boldsymbol{x}_t)$ using the following lemma inspired by proof approach of Liang et al. (2024b):

**Lemma 3** (Tilting factor). *For a fixed $\boldsymbol{x}_t$ we have*

$$q_{t-1|t}(\boldsymbol{x}_{t-1}|\boldsymbol{x}_t) \propto p^s_{t-1|t}(\boldsymbol{x}_{t-1}|\boldsymbol{x}_t)\exp\big(\zeta_{t,t-1}(\boldsymbol{x}_t, \boldsymbol{x}_{t-1})\big)$$

*with $\zeta_{t,t-1}(\boldsymbol{x}_t, \boldsymbol{x}_{t-1}) := \log q_{t-1}(\boldsymbol{x}_{t-1}) - \sqrt{\alpha_t}\boldsymbol{x}^T_{t-1}\nabla \log q^s_t(\boldsymbol{x}_t) + f(\boldsymbol{x}_t)$, where $f(\boldsymbol{x}_t)$ is an arbitrary function of $\boldsymbol{x}_t$.*

The notation $\propto$ in the Lemma 3 means proportional. Lemma 3 is employed in the proof of our Lemma 1 and its detailed proof is presented in Section B.5.

**Lemma 4** (Log-density of the backward process). *We have*

$$\sum_{t=1}^{T} \mathbb{E}_{X_t, X_{t-1} \sim \mathbb{Q}_{t,t-1}} \left[ \log \frac{p^s_{t-1|t}(X_{t-1}|X_t)}{\hat{p}_{t-1|t}(X_{t-1}|X_t)} \right]$$

$$\leq \sum_{t=1}^{T}(1-\alpha_t)\Bigg(\sqrt{\mathbb{E}_{X_t \sim Q_t}\|\nabla \log q_t(X_t) - \nabla \log q^s_t(X_t)\|^2 \mathbb{E}_{X_t \sim Q_t}\|\nabla \log q^s_t(X_t) - \hat{\kappa}\boldsymbol{s}_{\widehat{\Theta}_{\ell_1}}(X_t, t)\|^2}$$

$$+ \frac{1}{2}\mathbb{E}_{X_t \sim Q_t}\|\hat{\kappa}\boldsymbol{s}_{\widehat{\Theta}_{\ell_1}}(X_t, t) - \nabla \log q^s_t(X_t)\|^2\Bigg).$$

Lemma 4 is employed in the proof our main Theorem 1 and its detailed proof is presented in Section B.6.

### B.2 PROOF OF THEOREM 1

*Proof.* The poof approach is based on decomposing the total error using the Markov property of the forward and the backward process. We then, employ our Lemma 1 and Lemma 2 to handle individual terms.

Following the proof approach proposed by Liang et al. (2024b, Equation 13) (based on Markov property of the forward and the backward process), we can decompose the total error as

$$D_{\mathrm{KL}}(\mathbb{Q}_0||\widehat{P}_0) \leq D_{\mathrm{KL}}(\mathbb{Q}_T||\widehat{P}_T) + \sum_{t=1}^{T} \mathbb{E}_{X_t \sim \mathbb{Q}_t} \left[ D_{\mathrm{KL}}\big(\mathbb{Q}_{t-1|t}(.|X_t)||\widehat{P}_{t-1|t}(.|X_t)\big) \right].$$

We then employ the auxiliary function $q_t^s(\cdot)$ that satisfies our Assumption 4 (it doesn't need to be necessarily unique). We also use the notation $p_t^s(\cdot)$ for the reverse counterpart of the auxiliary function $q_t^s(\cdot)$ (see equation 7). We then rewrite the previous display employing $p_t^s(\cdot)$ and the definition of $D_{\mathrm{KL}}$:

$$D_{\mathrm{KL}}(\mathbb{Q}_0||\widehat{P}_0) \leq D_{\mathrm{KL}}(\mathbb{Q}_T||\widehat{P}_T) + \sum_{t=1}^{T} \mathbb{E}_{X_t \sim \mathbb{Q}_t} \left[ D_{\mathrm{KL}}\big(\mathbb{Q}_{t-1|t}(.|X_t)||\widehat{P}_{t-1|t}(.|X_t)\big) \right]$$

$$= \underbrace{D_{\mathrm{KL}}(\mathbb{Q}_T||\widehat{P}_T)}_{\text{Term 1: Initialization error}} + \underbrace{\sum_{t=1}^{T} \mathbb{E}_{X_t, X_{t-1} \sim \mathbb{Q}_{t,t-1}} \left[ \log \frac{q_{t-1|t}(X_{t-1}|X_t)}{p_{t-1|t}^s(X_{t-1}|X_t)} \right]}_{\text{Term 2: Reverse-step error}}$$

$$+ \underbrace{\sum_{t=1}^{T} \mathbb{E}_{X_t, X_{t-1} \sim \mathbb{Q}_{t,t-1}} \left[ \log \frac{p_{t-1|t}^s(X_{t-1}|X_t)}{\hat{p}_{t-1|t}(X_{t-1}|X_t)} \right]}_{\text{Term 3: Estimation error}}.$$

We now need to study each term specified above individually.

*Term 1: Initialization error*

Under the Assumption 1 and the assumed step size in equation 4, we have with Liang et al. (2024b, Remark 1)

$$D_{\mathrm{KL}}(\mathbb{Q}_T||\widehat{P}_T) \leq \frac{M}{T^2},$$

As stated by (Liang et al., 2024a, Definition 1), there exist cases that verify our step-size assumption, which in turn implies that $\bar{\alpha}_T \leq c/T^2$ for a constant $c \in (0, \infty)$.

*Term 2: Reverse-step error*

Under our Assumption 4 and Assumption 3 and with Lemma 1 we have

$$\sum_{t=1}^{T} \mathbb{E}_{X_t, X_{t-1} \sim \mathbb{Q}_{t,t-1}} \left[ \log \frac{q_{t-1|t}(X_{t-1}|X_t)}{p_{t-1|t}^s(X_{t-1}|X_t)} \right]$$

$$\leq \frac{1}{T} \big( s^2 B^2 + s^2 B^2 + s^2 B^2 \epsilon + sB\epsilon \big) + \Delta_T(\log q, \log q^s).$$

*Term 3: Estimation error*

We employ 1. Lemma 4, 2. adding a zero-valued term, 3. triangle inequality, and 4. the definition of our estimator and some linear algebra to obtain

$$\sum_{t=1}^{T} \mathbb{E}_{X_t, X_{t-1} \sim \mathbb{Q}_{t,t-1}} \left[ \log \frac{p_{t-1|t}^s(X_{t-1}|X_t)}{\hat{p}_{t-1|t}(X_{t-1}|X_t)} \right]$$

$$\leq \sum_{t=1}^{T} \frac{1 - \alpha_t}{2} \mathbb{E}_{X_t \sim Q_t} \| \hat{\kappa} \boldsymbol{s}_{\widehat{\Theta}_{\ell_1}}(X_t, t) - \nabla \log q_t^s(X_t) \|^2$$

$$+ \sum_{t=1}^{T}(1-\alpha_t)\sqrt{\mathbb{E}_{X_t \sim Q_t}\|\nabla \log q_t(X_t) - \nabla \log q_t^s(X_t)\|^2 \mathbb{E}_{X_t \sim Q_t}\|\nabla \log q_t^s(X_t) - \hat{\kappa}\boldsymbol{s}_{\widehat{\Theta}_{\ell_1}}(X_t, t)\|^2}$$

$$= \sum_{t=1}^{T}\frac{1-\alpha_t}{2}\left(\frac{1}{n}\sum_{i=1}^{n}\|\hat{\kappa}\boldsymbol{s}_{\widehat{\Theta}_{\ell_1}}(\boldsymbol{x}_t^i, t) - \nabla \log q_t^s(\boldsymbol{x}_t^i)\|^2 + \mathbb{E}_{X_t \sim Q_t}\|\hat{\kappa}\boldsymbol{s}_{\widehat{\Theta}_{\ell_1}}(X_t, t) - \nabla \log q_t^s(X_t)\|^2\right.$$

$$\left. - \frac{1}{n}\sum_{i=1}^{n}\|\hat{\kappa}\boldsymbol{s}_{\widehat{\Theta}_{\ell_1}}(\boldsymbol{x}_t^i, t) - \nabla \log q_t^s(\boldsymbol{x}_t^i)\|^2\right)$$

$$+ \sum_{t=1}^{T}(1-\alpha_t)\sqrt{\mathbb{E}_{X_t \sim Q_t}\|\nabla \log q_t(X_t) - \nabla \log q_t^s(X_t)\|^2 \mathbb{E}_{X_t \sim Q_t}\|\nabla \log q_t^s(X_t) - \hat{\kappa}\boldsymbol{s}_{\widehat{\Theta}_{\ell_1}}(X_t, t)\|^2}$$

$$\leq \sum_{t=1}^{T}\frac{1-\alpha_t}{2}\left(\frac{1}{n}\sum_{i=1}^{n}\|\hat{\kappa}\boldsymbol{s}_{\widehat{\Theta}_{\ell_1}}(\boldsymbol{x}_t^i, t) - \nabla \log q_t(\boldsymbol{x}_t^i)\|^2 + \frac{1}{n}\sum_{i=1}^{n}\|\nabla \log q_t(\boldsymbol{x}_t^i) - \nabla \log q_t^s(\boldsymbol{x}_t^i)\|^2\right.$$

$$\left. + \mathbb{E}_{X_t \sim Q_t}\|\hat{\kappa}\boldsymbol{s}_{\widehat{\Theta}_{\ell_1}}(X_t, t) - \nabla \log q_t^s(X_t)\|^2 - \frac{1}{n}\sum_{i=1}^{n}\|\hat{\kappa}\boldsymbol{s}_{\widehat{\Theta}_{\ell_1}}(\boldsymbol{x}_t^i, t) - \nabla \log q_t^s(\boldsymbol{x}_t^i)\|^2\right)$$

$$+ \sum_{t=1}^{T}(1-\alpha_t)\sqrt{\mathbb{E}_{X_t \sim Q_t}\|\nabla \log q_t(X_t) - \nabla \log q_t^s(X_t)\|^2 \mathbb{E}_{X_t \sim Q_t}\|\nabla \log q_t^s(X_t) - \hat{\kappa}\boldsymbol{s}_{\widehat{\Theta}_{\ell_1}}(X_t, t)\|^2}$$

$$\leq \sum_{t=1}^{T}\frac{1-\alpha_t}{2}\left(\frac{1}{n}\sum_{i=1}^{n}\|\kappa\boldsymbol{s}_{\Theta}(\boldsymbol{x}_t^i, t) - \nabla \log q_t(\boldsymbol{x}_t^i)\|^2 + r\kappa^2 - r\hat{\kappa}^2 + \frac{1}{n}\sum_{i=1}^{n}\|\nabla \log q_t(\boldsymbol{x}_t^i) - \nabla \log q_t^s(\boldsymbol{x}_t^i)\|^2\right.$$

$$\left. + \left|\mathbb{E}_{X_t \sim Q_t}\|\hat{\kappa}\boldsymbol{s}_{\widehat{\Theta}_{\ell_1}}(X_t, t) - \nabla \log q_t^s(X_t)\|^2 - \frac{1}{n}\sum_{i=1}^{n}\|\hat{\kappa}\boldsymbol{s}_{\widehat{\Theta}_{\ell_1}}(\boldsymbol{x}_t^i, t) - \nabla \log q_t^s(\boldsymbol{x}_t^i)\|^2\right|\right)$$

$$+ \sum_{t=1}^{T}(1-\alpha_t)\sqrt{\mathbb{E}_{X_t \sim Q_t}\|\nabla \log q_t(X_t) - \nabla \log q_t^s(X_t)\|^2 \mathbb{E}_{X_t \sim Q_t}\|\nabla \log q_t^s(X_t) - \hat{\kappa}\boldsymbol{s}_{\widehat{\Theta}_{\ell_1}}(X_t, t)\|^2}$$

for arbitrary function $\boldsymbol{s}_{\Theta}(\cdot, \cdot)$ with $\Theta \in \mathcal{B}_1$ and $\kappa \in (0, \infty)$.

Now, by collecting all the pieces of the proof we obtain

$$D_{\mathrm{KL}}(\mathbb{Q}_0 || \widehat{P}_0) \leq \frac{M}{T^2} + \frac{1}{T}\left(s^2 B^2 + s^2 B^2 + s^2 B^2 \epsilon + sB\epsilon\right) + \Delta_T(\log q, \log q^s)$$

$$+ \sum_{t=1}^{T}\frac{1-\alpha_t}{2}\left(\frac{1}{n}\sum_{i=1}^{n}\|\kappa\boldsymbol{s}_{\Theta}(\boldsymbol{x}_t^i, t) - \nabla \log q_t(\boldsymbol{x}_t^i)\|^2 + r\kappa^2 - r\hat{\kappa}^2\right.$$

$$+ \frac{1}{n}\sum_{i=1}^{n}\|\nabla \log q_t(\boldsymbol{x}_t^i) - \nabla \log q_t^s(\boldsymbol{x}_t^i)\|^2$$

$$+ \sup_{\Theta \in \mathcal{B}_1}\left|\mathbb{E}_{X_t \sim Q_t}\|\hat{\kappa}\boldsymbol{s}_{\Theta}(X_t, t) - \nabla \log q_t^s(X_t)\|^2 - \frac{1}{n}\sum_{i=1}^{n}\|\hat{\kappa}\boldsymbol{s}_{\Theta}(\boldsymbol{x}_t^i, t) - \nabla \log q_t^s(\boldsymbol{x}_t^i)\|^2\right|$$

$$\left. + 2\sqrt{\mathbb{E}_{X_t \sim Q_t}\|\nabla \log q_t(X_t) - \nabla \log q_t^s(X_t)\|^2 \mathbb{E}_{X_t \sim Q_t}\|\nabla \log q_t^s(X_t) - \hat{\kappa}\boldsymbol{s}_{\widehat{\Theta}_{\ell_1}}(X_t, t)\|^2}\right).$$

A high level idea now is choosing the tuning parameter $r$ in such a way that the term $-r\hat{\kappa}^2$ can dominate the terms in the absolute value that are dependent over $\hat{\kappa}$. The point here is that the terms in the absolute value are growing in the sparse function space. Employing Lemma 2 we obtain for $r \geq C_{\boldsymbol{x}}\sqrt{\log(np)/n}$ that

$$D_{\mathrm{KL}}(\mathbb{Q}_0 || \widehat{P}_0) \leq \frac{M}{T^2} + \frac{1}{T}\left(s^2 B^2 + s^2 B^2 + s^2 B^2 \epsilon + sB\epsilon\right) + \Delta_T(\log q, \log q^s)$$

$$+ \inf_{\Theta \in \mathcal{B}_1; \kappa \in (0, \infty)}\left\{\sum_{t=1}^{T}\frac{1-\alpha_t}{2}\frac{1}{n}\sum_{i=1}^{n}\|\kappa\boldsymbol{s}_{\Theta}(\boldsymbol{x}_t^i, t) - \nabla \log q_t(\boldsymbol{x}_t^i)\|^2 + r\kappa^2\right\}$$

$$+ \sum_{t=1}^{T} \frac{1-\alpha_t}{2} \left( \frac{1}{n} \sum_{i=1}^{n} \|\nabla \log q_t(\boldsymbol{x}_t^i) - \nabla \log q_t^s(\boldsymbol{x}_t^i)\|^2 + C_x s^2 B^2 \frac{\sqrt{\log(np)}}{\sqrt{n}} \right.$$

$$\left. + 2\sqrt{\mathbb{E}_{X_t \sim Q_t} \|\nabla \log q_t(X_t) - \nabla \log q_t^s(X_t)\|^2 \mathbb{E}_{X_t \sim Q_t} \|\nabla \log q_t^s(X_t) - \hat{\kappa} \boldsymbol{s}_{\widehat{\Theta}_{\ell_1}}(X_t,t)\|^2} \right)$$

with probability at least $1 - 32/n$.

Using Assumption 4, equation 4, and equation 2 we also obtain

$$\sum_{t=1}^{T} (1-\alpha_t) \sqrt{\mathbb{E}_{X_t \sim Q_t} \|\nabla \log q_t(X_t) - \nabla \log q_t^s(X_t)\|^2 \mathbb{E}_{X_t \sim Q_t} \|\nabla \log q_t^s(X_t) - \hat{\kappa} \boldsymbol{s}_{\widehat{\Theta}_{\ell_1}}(X_t,t)\|^2}$$

$$\leq \left( \max_{t \in \{1,\dots,T\}} \sqrt{\mathbb{E}_{X_t \sim Q_t} \|\nabla \log q_t^s(X_t) - \hat{\kappa} \boldsymbol{s}_{\widehat{\Theta}_{\ell_1}}(X_t,t)\|^2} \right)$$

$$\sum_{t=1}^{T} (1-\alpha_t) \sqrt{\mathbb{E}_{X_t \sim Q_t} \|\nabla \log q_t(X_t) - \nabla \log q_t^s(X_t)\|^2}$$

$$\leq \epsilon \left( \max_{t \in \{1,\dots,T\}} \sqrt{\mathbb{E}_{X_t \sim Q_t} \left( 2\|\nabla \log q_t^s(X_t)\|^2 + 2\|\hat{\kappa} \boldsymbol{s}_{\widehat{\Theta}_{\ell_1}}(X_t,t)\|^2 \right)} \right)$$

$$\leq \epsilon \left( \sqrt{2(sB)^2 + 2\hat{\kappa}^2} \right)$$

$$\leq 5\epsilon sB$$

$$\leq 5\frac{sB}{T},$$

where for the fourth inequality we can follow the same approach as in Taheri et al. (2021, Page 155) to conclude that for large enough tuning (just double it), $\hat{\kappa} \leq 3\kappa^*$ (where $\kappa^* \approx sB$) that gives us the space to remove $\hat{\kappa}$ from our bounds. Let's also note that under the Assumption 4 and equation 4, we can also conclude that

$$\sum_{t=1}^{T} (1-\alpha_t) \frac{1}{n} \sum_{i=1}^{n} \|\nabla \log q_t(\boldsymbol{x}_t^i) - \nabla \log q_t^s(\boldsymbol{x}_t^i)\|^2 \lesssim \epsilon \leq \frac{1}{T}.$$

Finally, collecting displays above, some simplifications, and keeping the dominant factors gives us the desired results

$$D_{\mathrm{KL}}(\mathbb{Q}_0 \| \widehat{P}_0) \leq \frac{M}{T^2} + \frac{1}{T} \left( s^2 B^2 + s^2 B^2 + s^2 B^2 \epsilon + sB\epsilon \right) + \Delta_T(\log q, \log q^s)$$

$$+ \inf_{\Theta \in \mathcal{B}_1; \kappa \in (0,\infty)} \left\{ \sum_{t=1}^{T} \frac{1-\alpha_t}{2} \frac{1}{n} \sum_{i=1}^{n} \|\kappa \boldsymbol{s}_{\Theta}(\boldsymbol{x}_t^i, t) - \nabla \log q_t(\boldsymbol{x}_t^i)\|^2 + r\kappa^2 \right\}$$

$$+ \frac{1}{T} + s^2 B^2 \frac{\sqrt{\log(np)}}{\sqrt{n}} + 5\frac{sB}{T}$$

$$\leq \frac{M}{T^2} + \frac{1}{T} \max\{1, 9(sB)^2\} + C_{\boldsymbol{x}} s^2 B^2 \frac{\sqrt{\log(nTp)}}{\sqrt{n}} + \Delta_T(\log q, \log q^s)$$

$$+ \inf_{\substack{\Theta \in \mathcal{B}_1 \\ \kappa \in (0,\infty)}} \left\{ \frac{\log T}{T} \sum_{t=1}^{T} \frac{1}{n} \sum_{i=1}^{n} \|\kappa \boldsymbol{s}_{\Theta}(\boldsymbol{x}_t^i, t) - \nabla_{\boldsymbol{x}_t} \log q_t(\boldsymbol{x}_t^i)\|^2 + r\kappa^2 \right\}.$$

$$\square$$

### B.3 PROOF OF LEMMA 1

*Proof.* We start the proof with Lemma 3 that relates $q_{t-1|t}(\boldsymbol{x}_{t-1}|\boldsymbol{x}_t)$ and $p_{t-1|t}^s(\boldsymbol{x}_{t-1}|\boldsymbol{x}_t)$ by

$$q_{t-1|t}(\boldsymbol{x}_{t-1}|\boldsymbol{x}_t) \propto p_{t-1|t}^s(\boldsymbol{x}_{t-1}|\boldsymbol{x}_t) \exp\left(\zeta_{t,t-1}(\boldsymbol{x}_t, \boldsymbol{x}_{t-1})\right) \tag{8}$$

with $\zeta_{t,t-1}(\boldsymbol{x}_t, \boldsymbol{x}_{t-1}) = \log q_{t-1}(\boldsymbol{x}_{t-1}) - \sqrt{\alpha_t} \boldsymbol{x}_{t-1}^T \nabla \log q_t^s(\boldsymbol{x}_t) + f(\boldsymbol{x}_t)$, where $f(\boldsymbol{x}_t)$ is an arbitrary function of $\boldsymbol{x}_t$. Now, let's progress with adding a zero-valued term to the $\zeta_{t,t-1}(\boldsymbol{x}_t, \boldsymbol{x}_{t-1})$

$$\zeta_{t,t-1}(\boldsymbol{x}_t, \boldsymbol{x}_{t-1}) = \log q_{t-1}^s(\boldsymbol{x}_{t-1}) - \sqrt{\alpha_t}\boldsymbol{x}_{t-1}^T \nabla \log q_t^s(\boldsymbol{x}_t)$$
$$+ \log q_{t-1}(\boldsymbol{x}_{t-1}) - \log q_{t-1}^s(\boldsymbol{x}_{t-1}) + f(\boldsymbol{x}_t)$$

and set $f(\boldsymbol{x}_t) = -\log q_{t-1}^s(\boldsymbol{u}_t^s) + \sqrt{\alpha_t}(\boldsymbol{u}_t^s)^T \nabla \log q_t^s(\boldsymbol{x}_t)$. Then, we have

$$\zeta_{t,t-1}(\boldsymbol{x}_t, \boldsymbol{x}_{t-1}) = \log q_{t-1}^s(\boldsymbol{x}_{t-1}) - \log q_{t-1}^s(\boldsymbol{u}_t^s) - (\boldsymbol{x}_{t-1} - \boldsymbol{u}_t^s)^T \sqrt{\alpha_t} \nabla \log q_t^s(\boldsymbol{x}_t)$$
$$+ \log q_{t-1}(\boldsymbol{x}_{t-1}) - \log q_{t-1}^s(\boldsymbol{x}_{t-1})\,.$$

For a fixed $\boldsymbol{x}_t$, then we have (the denominator is for normalization reason)

$$q_{t-1|t}(\boldsymbol{x}_{t-1}|\boldsymbol{x}_t) = \frac{p_{t-1|t}^s(\boldsymbol{x}_{t-1}|\boldsymbol{x}_t) \exp\big(\zeta_{t,t-1}(\boldsymbol{x}_t, \boldsymbol{x}_{t-1})\big)}{\mathbb{E}_{X_{t-1} \sim P_{t-1|t}^s}\big[\exp\big(\zeta_{t,t-1}(\boldsymbol{x}_t, X_{t-1})\big)\big]}\,.$$

We then use the above display and Jensen's inequality to obtain

$$\mathbb{E}_{X_t, X_{t-1} \sim \mathbb{Q}_{t,t-1}}\left[\log \frac{q_{t-1|t}(X_{t-1}|X_t)}{p_{t-1|t}^s(X_{t-1}|X_t)}\right]$$

$$= \mathbb{E}_{X_t, X_{t-1} \sim \mathbb{Q}_{t,t-1}}\left[\zeta_{t,t-1}(X_t, X_{t-1}) - \log \mathbb{E}_{X_{t-1} \sim P_{t-1|t}^s}\big[\exp\big(\zeta_{t,t-1}(X_t, X_{t-1})\big)\big]\right]$$

$$= \mathbb{E}_{X_t, X_{t-1} \sim \mathbb{Q}_{t,t-1}}\big[\zeta_{t,t-1}(X_t, X_{t-1})\big] - \mathbb{E}_{X_t \sim \mathbb{Q}_t}\left[\log \mathbb{E}_{X_{t-1} \sim P_{t-1|t}^s}\big[\exp\big(\zeta_{t,t-1}(X_t, X_{t-1})\big)\big]\right]$$

$$\leq \mathbb{E}_{X_t, X_{t-1} \sim \mathbb{Q}_{t,t-1}}\big[\zeta_{t,t-1}(X_t, X_{t-1})\big] - \mathbb{E}_{X_t \sim \mathbb{Q}_t}\left[\mathbb{E}_{X_{t-1} \sim P_{t-1|t}^s}\big[\zeta_{t,t-1}(X_t, X_{t-1})\big]\right]\,.$$

Now, let's rewrite

$$\zeta_{t,t-1}(\boldsymbol{x}_t, \boldsymbol{x}_{t-1}) = \log q_{t-1}^s(\boldsymbol{x}_{t-1}) - \log q_{t-1}^s(\boldsymbol{u}_t^s) - (\boldsymbol{x}_{t-1} - \boldsymbol{u}_t^s)^T \sqrt{\alpha_t} \nabla \log q_t^s(\boldsymbol{x}_t)$$
$$+ \log q_{t-1}(\boldsymbol{x}_{t-1}) - \log q_{t-1}^s(\boldsymbol{x}_{t-1})$$
$$=: \zeta_{t,t-1}'(\boldsymbol{x}_t, \boldsymbol{x}_{t-1}) + \log q_{t-1}(\boldsymbol{x}_{t-1}) - \log q_{t-1}^s(\boldsymbol{x}_{t-1})\,.$$

We are now left with three terms: 1. $\mathbb{E}_{X_t, X_{t-1} \sim \mathbb{Q}_{t,t-1}}[\zeta_{t,t-1}'(X_t, X_{t-1})]$, 2. $\mathbb{E}_{X_t \sim \mathbb{Q}_t, X_{t-1} \sim P_{t-1|t}^s}[\zeta_{t,t-1}'(X_t, X_{t-1})]$, and 3. $\mathbb{E}_{X_t \sim \mathbb{Q}_t}[\mathbb{E}_{X_{t-1} \sim \mathbb{Q}_{t-1|t}}[\log q_{t-1}(X_{t-1}) - \log q_{t-1}^s(X_{t-1})] - \mathbb{E}_{X_{t-1} \sim P_{t-1|t}^s}[\log q_{t-1}(X_{t-1}) - \log q_{t-1}^s(X_{t-1})]]$ and we need to study 1. and 2. in details:

*Term 1:* $\mathbb{E}_{X_t \sim \mathbb{Q}_t, X_{t-1} \sim P_{t-1|t}^s}[\zeta_{t,t-1}'(X_t, X_{t-1})]$

We 1. use the definition of $\zeta_{t,t-1}'(X_t, X_{t-1})$, 2. (Second order) Taylor expand $\log q_{t-1}^s(X_{t-1})$ around $\boldsymbol{u}_t^s$, 3. use Liang et al. (2024b, Lemma 7) that implies $\mathbb{E}_{X_{t-1} \sim P_{t-1|t}^s}[(X_{t-1}^i - (\boldsymbol{u}^s)_t^i)^p] = 0 \ \forall p \geq 1 \ odd$ (we use the notation $X_t^i$ to referenec to the $i$th feature of the vector $X_t$), 4.using the fact that $X_{t-1}^i$ is conditionally independent of $X_{t-1}^j$ for $i \neq j$ and again $\mathbb{E}_{X_{t-1} \sim P_{t-1|t}^s}[(X_{t-1}^i - (\boldsymbol{u}^s)_t^i)^p] = 0 \ \forall p \geq 1 \ odd$, and 5. $\mathbb{E}_{X_{t-1} \sim P_{t-1|t}^s}[(X_{t-1}^i - (\boldsymbol{u}^s)_t^i)^2] = (1 - \alpha_t)/\alpha_t$ (see equation 7)

$$\mathbb{E}_{X_t \sim \mathbb{Q}_t, X_{t-1} \sim P_{t-1|t}^s}[\zeta_{t,t-1}'(X_t, X_{t-1})]$$

$$= \mathbb{E}_{X_t \sim \mathbb{Q}_t, X_{t-1} \sim P_{t-1|t}^s}\big[\log q_{t-1}^s(X_{t-1}) - \log q_{t-1}^s(\boldsymbol{u}_t^s) - (X_{t-1} - \boldsymbol{u}_t^s)^T \sqrt{\alpha_t} \nabla \log q_t^s(X_t)\big]$$

$$\approx \mathbb{E}_{X_t \sim \mathbb{Q}_t, X_{t-1} \sim P_{t-1|t}^s}\Big[\nabla \log q_{t-1}^s(\boldsymbol{u}_t^s)(X_{t-1} - \boldsymbol{u}_t^s) - (X_{t-1} - \boldsymbol{u}_t^s)^T \sqrt{\alpha_t} \nabla \log q_t^s(X_t)$$
$$+ \frac{1}{2}(X_{t-1} - \boldsymbol{u}_t^s)^T \nabla^2 \log q_{t-1}^s(\boldsymbol{u}_t^s)(X_{t-1} - \boldsymbol{u}_t^s)\Big]$$

$$= \mathbb{E}_{X_t \sim \mathbb{Q}_t, X_{t-1} \sim P_{t-1|t}^s}\Big[\frac{1}{2}(X_{t-1} - \boldsymbol{u}_t^s)^T \nabla^2 \log q_{t-1}^s(\boldsymbol{u}_t^s)(X_{t-1} - \boldsymbol{u}_t^s)\Big]$$

$$= \frac{1}{2}\sum_{i=1}^d \mathbb{E}_{X_t \sim \mathbb{Q}_t}\big[\big(\nabla^2 \log q_{t-1}^s(\boldsymbol{u}_t^s)\big)_{ii} \mathbb{E}_{X_{t-1} \sim P_{t-1|t}^s}\big(X_{t-1}^i - (\boldsymbol{u}_t^s)^i\big)^2\big]$$

$$= \frac{(1 - \alpha_t)}{2\alpha_t}\sum_{i=1}^d \mathbb{E}_{X_t \sim \mathbb{Q}_t}\big[\big(\nabla^2 \log q_{t-1}^s(\boldsymbol{u}_t^s)\big)_{ii}\big]\,.$$

Note that here (as well as for the Term 2), for keeping the proofs simple and tractable, we use a second-order Taylor expansion and employ the notation $\approx$. However, higher-order expansions can also be applied without affecting the dominant rates. As we extend to higher-order Taylor expansions, the dominant factor remains $O((1-\alpha_t)/\alpha_t)^2$, so we omit those terms for simplicity.

*Term 2:* $\mathbb{E}_{X_t,X_{t-1}\sim\mathbb{Q}_{t,t-1}}[\zeta'_{t,t-1}(X_t,X_{t-1})]$

Following the same approach as in previous step and some further linear algebra we obtain

$$
\begin{aligned}
&\mathbb{E}_{X_t,X_{t-1}\sim\mathbb{Q}_{t,t-1}}\big[\zeta'_{t,t-1}(X_t,X_{t-1})\big]\\
&= \mathbb{E}_{X_t,X_{t-1}\sim\mathbb{Q}_{t,t-1}}\big[\log q^s_{t-1}(X_{t-1}) - \log q^s_{t-1}(\boldsymbol{u}^s_t) - (X_{t-1}-\boldsymbol{u}^s_t)^T\sqrt{\alpha_t}\nabla\log q^s_t(X_t)\big]\\
&\approx \mathbb{E}_{X_t,X_{t-1}\sim\mathbb{Q}_{t,t-1}}\Big[\nabla\log q^s_{t-1}(\boldsymbol{u}^s_t)(X_{t-1}-\boldsymbol{u}^s_t) - (X_{t-1}-\boldsymbol{u}^s_t)^T\sqrt{\alpha_t}\nabla\log q^s_t(X_t)\\
&\quad + \frac{1}{2}(X_{t-1}-\boldsymbol{u}^s_t)^T\nabla^2\log q^s_{t-1}(\boldsymbol{u}^s_t)(X_{t-1}-\boldsymbol{u}^s_t)\Big]\\
&= (1-\sqrt{\alpha_t})\mathbb{E}_{X_t\sim\mathbb{Q}_t}\big[\nabla\log q^s_{t-1}(\boldsymbol{u}^s_t)\mathbb{E}_{X_{t-1}\sim\mathbb{Q}_{t-1|t}}[(X_{t-1}-\boldsymbol{u}^s_t)]\big]\\
&\quad + \frac{1}{2}\mathbb{E}_{X_t,X_{t-1}\sim\mathbb{Q}_{t,t-1}}\big[(X_{t-1}-\boldsymbol{u}^s_t)^T\nabla^2\log q^s_{t-1}(\boldsymbol{u}^s_t)(X_{t-1}-\boldsymbol{u}^s_t)\big].
\end{aligned}
$$

Employing Liang et al. (2024b, Lemma 8; first claim), we have $\mathbb{E}_{X_{t-1}\sim\mathbb{Q}_{t-1|t}}[X_{t-1}] = \boldsymbol{u}_t$. That implies

$$
\begin{aligned}
\mathbb{E}_{X_{t-1}\sim\mathbb{Q}_{t-1|t}}[(X_{t-1}-\boldsymbol{u}^s_t)] &= \mathbb{E}_{X_{t-1}\sim\mathbb{Q}_{t-1|t}}[X_{t-1}] - \boldsymbol{u}^s_t\\
&= \boldsymbol{u}_t - \frac{1}{\sqrt{\alpha_t}}\big(\boldsymbol{x}_t + (1-\alpha_t)\nabla\log q^s_t(\boldsymbol{x}_t)\big)\\
&= \frac{(1-\alpha_t)}{\sqrt{\alpha_t}}\big(\nabla\log q_t(\boldsymbol{x}_t) - \nabla\log q^s_t(\boldsymbol{x}_t)\big).
\end{aligned}
$$

Collecting the pieces above together with Cauchy–Schwarz inequality we obtain

$$
\begin{aligned}
&\mathbb{E}_{X_t,X_{t-1}\sim\mathbb{Q}_{t,t-1}}\big[\zeta'_{t,t-1}(X_t,X_{t-1})\big]\\
&\leq \frac{(1-\alpha_t)}{\sqrt{\alpha_t}}(1-\sqrt{\alpha_t})\sqrt{\mathbb{E}_{X_t\sim\mathbb{Q}_t}\|\nabla\log q^s_{t-1}(\boldsymbol{u}^s_t)\|^2\mathbb{E}_{X_t\sim\mathbb{Q}_t}\|\nabla\log q_t(X_t)-\nabla\log q^s_t(X_t)\|^2}\\
&\quad + \frac{1}{2}\mathbb{E}_{X_t,X_{t-1}\sim\mathbb{Q}_{t,t-1}}\big[(X_{t-1}-\boldsymbol{u}^s_t)^T\nabla^2\log q^s_{t-1}(\boldsymbol{u}^s_t)(X_{t-1}-\boldsymbol{u}^s_t)\big].
\end{aligned}
$$

Now let treat the second term in the inequality above by 1. adding a zero-valued term, 2. expanding the product, 3. using Liang et al. (2024b, Lemma 8; second claim) and the fact that terms two and three goes to zero and some rewriting

$$
\begin{aligned}
&\mathbb{E}_{X_t\sim\mathbb{Q}_t}\big[\mathbb{E}_{X_{t-1}\sim\mathbb{Q}_{t-1|t}}[(X_{t-1}-\boldsymbol{u}_t+\boldsymbol{u}_t-\boldsymbol{u}^s_t)^T(X_{t-1}-\boldsymbol{u}_t+\boldsymbol{u}_t-\boldsymbol{u}^s_t)]\big]\\
&= \mathbb{E}_{X_t\sim\mathbb{Q}_t}\big[\mathbb{E}_{X_{t-1}\sim\mathbb{Q}_{t-1|t}}[(X_{t-1}-\boldsymbol{u}_t)^T(X_{t-1}-\boldsymbol{u}_t)]\big]\\
&\quad + \mathbb{E}_{X_t\sim\mathbb{Q}_t}\big[\mathbb{E}_{X_{t-1}\sim\mathbb{Q}_{t-1|t}}[(X_{t-1}-\boldsymbol{u}_t)^T(\boldsymbol{u}_t-\boldsymbol{u}^s_t)]\big]\\
&\quad + \mathbb{E}_{X_t\sim\mathbb{Q}_t}\big[\mathbb{E}_{X_{t-1}\sim\mathbb{Q}_{t-1|t}}[(\boldsymbol{u}_t-\boldsymbol{u}^s_t)^T(X_{t-1}-\boldsymbol{u}_t)]\big]\\
&\quad + \mathbb{E}_{X_t\sim\mathbb{Q}_t}\big[\mathbb{E}_{X_{t-1}\sim\mathbb{Q}_{t-1|t}}[(\boldsymbol{u}_t-\boldsymbol{u}^s_t)^T(\boldsymbol{u}_t-\boldsymbol{u}^s_t)]\big]\\
&= \mathbb{E}_{X_t\sim\mathbb{Q}_t}\Big[\frac{1-\alpha_t}{\alpha_t}\mathbb{I}_d + \frac{(1-\alpha_t)^2}{\alpha_t}\nabla^2\log q_t(X_t)\Big]\\
&\quad + \mathbb{E}_{X_t\sim\mathbb{Q}_t}\Big[\frac{(1-\alpha_t)^2}{\alpha_t}\big(\nabla\log q^s_t(X_t)-\nabla\log q_t(X_t)\big)^T\big(\nabla\log q^s_t(X_t)-\nabla\log q_t(X_t)\big)\Big].
\end{aligned}
$$
(9)

Above results state that our term involving $(1-\alpha_t)\mathbb{I}_d/\alpha_t$ can be canceled out by the terms from Step 1. We then need to study the remaining terms that all involve the nice factor $(1-\alpha_t)^2$.

So, collecting all the pieces of the proof, we 1. use the results from Term 1. and Term 2., 2. implying some linear algebra to expand the product, 3. use equation 9 and cancel out terms involving the multiple $(1 - \alpha_t)$ (for simplicity we have ignored the last term in equation 9 since it has a minor affect on our final rates), 4. using our Assumption 4, $\mathbb{E}_{X_t \sim \mathbb{Q}_t} \|\nabla \log q_{t-1}^s(\boldsymbol{u}_t^s)\|^2 \leq s^2 B^2$, 5. once again use the Assumption 4 that implies $s$ sparsity between entries of $\nabla \log q_{t-1}^s(\boldsymbol{u}_t^s)$, that also implies sparsity for the second order derivative (it causes that just a fraction of entries get involved in those sums) (also note that the last term is appeared regarding the term that we neglected in equation 9), and 6. use our Assumption over the step sizes equation 4 to obtain

$$
\sum_{t=1}^{T} \mathbb{E}_{X_t, X_{t-1} \sim \mathbb{Q}_{t,t-1}} \left[ \log \frac{q_{t-1|t}(X_{t-1}|X_t)}{p_{t-1|t}^s(X_{t-1}|X_t)} \right]
$$

$$
\leq \sum_{t=1}^{T} \left( \frac{(1-\alpha_t)}{\sqrt{\alpha_t}}(1 - \sqrt{\alpha_t}) \sqrt{\mathbb{E}_{X_t \sim \mathbb{Q}_t} \|\nabla \log q_{t-1}^s(\boldsymbol{u}_t^s)\|^2 \mathbb{E}_{X_t \sim \mathbb{Q}_t} \|\nabla \log q_t(X_t) - \nabla \log q_t^s(X_t)\|^2} \right.
$$

$$
+ \frac{1}{2} \mathbb{E}_{X_t, X_{t-1} \sim \mathbb{Q}_{t,t-1}} \left[ (X_{t-1} - \boldsymbol{u}_t^s)^T \nabla^2 \log q_{t-1}^s(\boldsymbol{u}_t^s)(X_{t-1} - \boldsymbol{u}_t^s) \right]
$$

$$
\left. - \frac{1}{2} \sum_{i=1}^{d} \mathbb{E}_{X_t \sim \mathbb{Q}_t} \left( \nabla^2 \log q_{t-1}^s(\boldsymbol{u}_t^s) \right)_{ii} \left( \frac{1-\alpha_t}{\alpha_t} \right) \right)
$$

$$
+ \sum_{t=1}^{T} \left( \mathbb{E}_{X_t \sim \mathbb{Q}_t} [\mathbb{E}_{X_{t-1} \sim \mathbb{Q}_{t-1|t}} [\log q_{t-1}(X_{t-1}) - \log q_{t-1}^s(X_{t-1})] \right.
$$

$$
\left. - \mathbb{E}_{X_{t-1} \sim P_{t-1|t}^s} [\log q_{t-1}(X_{t-1}) - \log q_{t-1}^s(X_{t-1})]] \right)
$$

$$
\leq \sum_{t=1}^{T} \left( \frac{(1-\alpha_t)}{\sqrt{\alpha_t}}(1 - \sqrt{\alpha_t}) \sqrt{\mathbb{E}_{X_t \sim \mathbb{Q}_t} \|\nabla \log q_{t-1}^s(\boldsymbol{u}_t^s)\|^2 \mathbb{E}_{X_t \sim \mathbb{Q}_t} \|\nabla \log q_t(X_t) - \nabla \log q_t^s(X_t)\|^2} \right.
$$

$$
+ \frac{1}{2} \mathbb{E}_{X_t, X_{t-1} \sim \mathbb{Q}_{t,t-1}} \left[ \sum_{i=1}^{d} (X_{t-1} - \boldsymbol{u}_t^s)_i^2 \left( \nabla^2 \log q_{t-1}^s(\boldsymbol{u}_t^s) \right)_{ii} \right.
$$

$$
\left. + \sum_{i=1}^{d} \sum_{j=1, j \neq i}^{d} (X_{t-1} - \boldsymbol{u}_t^s)_i (X_{t-1} - \boldsymbol{u}_t^s)_j \left( \nabla^2 \log q_{t-1}^s(\boldsymbol{u}_t^s) \right)_{ij} \right]
$$

$$
\left. - \frac{1}{2} \sum_{i=1}^{d} \mathbb{E}_{X_t \sim \mathbb{Q}_t} \left( \nabla^2 \log q_{t-1}^s(\boldsymbol{u}_t^s) \right)_{ii} \left( \frac{1-\alpha_t}{\alpha_t} \right) \right)
$$

$$
+ \sum_{t=1}^{T} \left( \mathbb{E}_{X_t \sim \mathbb{Q}_t} [\mathbb{E}_{X_{t-1} \sim \mathbb{Q}_{t-1|t}} [\log q_{t-1}(X_{t-1}) - \log q_{t-1}^s(X_{t-1})] \right.
$$

$$
\left. - \mathbb{E}_{X_{t-1} \sim P_{t-1|t}^s} [\log q_{t-1}(X_{t-1}) - \log q_{t-1}^s(X_{t-1})]] \right)
$$

$$
\leq \sum_{t=1}^{T} \left( \frac{(1-\alpha_t)}{\sqrt{\alpha_t}}(1 - \sqrt{\alpha_t}) \sqrt{\mathbb{E}_{X_t \sim \mathbb{Q}_t} \|\nabla \log q_{t-1}^s(\boldsymbol{u}_t^s)\|^2 \mathbb{E}_{X_t \sim \mathbb{Q}_t} \|\nabla \log q_t(X_t) - \nabla \log q_t^s(X_t)\|^2} \right.
$$

$$
+ \frac{1}{2} \mathbb{E}_{X_t, X_{t-1} \sim \mathbb{Q}_{t,t-1}} \left[ \sum_{i=1}^{d} \frac{(1-\alpha_t)^2}{\alpha_t} \left( \nabla^2 \log q_t(X_t) \right)_{ii} \left( \nabla^2 \log q_{t-1}^s(\boldsymbol{u}_t^s) \right)_{ii} \right.
$$

$$
\left. \left. + \sum_{i=1}^{d} \sum_{j=1, j \neq i}^{d} \frac{(1-\alpha_t)^2}{\alpha_t} \left( \nabla^2 \log q_t(X_t) \right)_{ij} \left( \nabla^2 \log q_{t-1}^s(\boldsymbol{u}_t^s) \right)_{ij} \right] \right)
$$

$$
+ \sum_{t=1}^{T} \left( \mathbb{E}_{X_t \sim \mathbb{Q}_t} [\mathbb{E}_{X_{t-1} \sim \mathbb{Q}_{t-1|t}} [\log q_{t-1}(X_{t-1}) - \log q_{t-1}^s(X_{t-1})] \right.
$$

$$
\left. - \mathbb{E}_{X_{t-1} \sim P_{t-1|t}^s} [\log q_{t-1}(X_{t-1}) - \log q_{t-1}^s(X_{t-1})]] \right)
$$

$$\leq \sum_{t=1}^{T}\left(\frac{(1-\alpha_t)}{\sqrt{\alpha_t}}(1-\sqrt{\alpha_t})\sqrt{s^2 B^2 \mathbb{E}_{X_t \sim \mathbb{Q}_t}\|\nabla \log q_t(X_t) - \nabla \log q_t^s(X_t)\|^2}\right.$$

$$+\frac{1}{2}\mathbb{E}_{X_t, X_{t-1} \sim \mathbb{Q}_{t,t-1}}\left[\sum_{i=1}^{d}\frac{(1-\alpha_t)^2}{\alpha_t}\left(\nabla^2 \log q_t(X_t)\right)_{ii}\left(\nabla^2 \log q_{t-1}^s(\boldsymbol{u}_t^s)\right)_{ii}\right.$$

$$\left.\left.+\sum_{i=1}^{d}\sum_{j=1,j\neq i}^{d}\frac{(1-\alpha_t)^2}{\alpha_t}\left(\nabla^2 \log q_t(X_t)\right)_{ij}\left(\nabla^2 \log q_{t-1}^s(\boldsymbol{u}_t^s)\right)_{ij}\right]\right)$$

$$+\sum_{t=1}^{T}\left(\mathbb{E}_{X_t \sim \mathbb{Q}_t}[\mathbb{E}_{X_{t-1} \sim \mathbb{Q}_{t-1|t}}[\log q_{t-1}(X_{t-1}) - \log q_{t-1}^s(X_{t-1})]\right.$$

$$\left.-\mathbb{E}_{X_{t-1} \sim P_{t-1|t}^s}[\log q_{t-1}(X_{t-1}) - \log q_{t-1}^s(X_{t-1})]]\right)$$

$$\leq \sum_{t=1}^{T}\left(\frac{(1-\alpha_t)}{\sqrt{\alpha_t}}(1-\sqrt{\alpha_t})(sB\epsilon) + \frac{(1-\alpha_t)^2}{2\alpha_t}(s^2 B^2) + \frac{(1-\alpha_t)^2}{\alpha_t}(s^2 B^2) + \frac{(1-\alpha_t)^2}{\alpha_t}(s^2 B^2 \epsilon)\right)$$

$$+\sum_{t=1}^{T}\left(\mathbb{E}_{X_t \sim \mathbb{Q}_t}[\mathbb{E}_{X_{t-1} \sim \mathbb{Q}_{t-1|t}}[\log q_{t-1}(X_{t-1}) - \log q_{t-1}^s(X_{t-1})]\right.$$

$$\left.-\mathbb{E}_{X_{t-1} \sim P_{t-1|t}^s}[\log q_{t-1}(X_{t-1}) - \log q_{t-1}^s(X_{t-1})]]\right)$$

$$\leq \frac{sB\epsilon}{T} + \frac{s^2 B^2}{T} + \frac{s^2 B^2}{T} + \frac{s^2 B^2 \epsilon}{T}$$

$$+\sum_{t=1}^{T}\left(\mathbb{E}_{X_t \sim \mathbb{Q}_t}[\mathbb{E}_{X_{t-1} \sim \mathbb{Q}_{t-1|t}}[\log q_{t-1}(X_{t-1}) - \log q_{t-1}^s(X_{t-1})]\right.$$

$$\left.-\mathbb{E}_{X_{t-1} \sim P_{t-1|t}^s}[\log q_{t-1}(X_{t-1}) - \log q_{t-1}^s(X_{t-1})]]\right),$$

as desired.

$\square$

### B.4 PROOF OF LEMMA 2

*Proof.* Our proof approach is based on the tools from empirical process theory and our sparsity assumptions over the network space and $\nabla \log q_t^s(\boldsymbol{x}_t)$.

Let start with the application of symmetrization of probabilities with $\zeta_i$ for $i \in \{1, \ldots, n\}$ as i.i.d. Rademacher random variables that are independent of the data (van de Geer, 2016, Lemma 16.1), and employing Contrcation principle (Ledoux & Talagrand, 2013, Theorem 4.4) to obtain

$$\Pr\left(\sup_{\Theta \in \mathcal{B}_1}\left|\mathbb{E}_{X_t \sim Q_t}\|\hat{\kappa}\boldsymbol{s}_\Theta(X_t, t) - \nabla \log q_t^s(X_t, t)\|^2 - \frac{1}{n}\sum_{i=1}^{n}\|\hat{\kappa}\boldsymbol{s}_\Theta(\boldsymbol{x}_t^i, t) - \nabla \log q_t^s(\boldsymbol{x}_t^i)\|^2\right| \geq 4\mathcal{R}\sqrt{\frac{2t}{n}}\right)$$

$$\leq 4\Pr\left(\sup_{\Theta \in \mathcal{B}_1}\left|\frac{1}{n}\sum_{i=1}^{n}\zeta_i\|\hat{\kappa}\boldsymbol{s}_\Theta(\boldsymbol{x}_t^i, t) - \nabla \log q_t^s(\boldsymbol{x}_t^i)\|^2\right| \geq \mathcal{R}\sqrt{\frac{2t}{n}}\right)$$

$$\leq 8\Pr\left(\sup_{\Theta \in \mathcal{B}_1}\left|\frac{1}{n}\sum_{i=1}^{n}2\zeta_i\left(\|\hat{\kappa}\boldsymbol{s}_\Theta(\boldsymbol{x}_t^i, t)\|^2 + \|\nabla \log q_t^s(\boldsymbol{x}_t^i)\|^2\right)\right| \geq \mathcal{R}\sqrt{\frac{2t}{n}}\right)$$

$$\leq 8\Pr\left(\sup_{\Theta \in \mathcal{B}_1}\left|\frac{1}{n}\sum_{i=1}^{n}2\zeta_i\left(\|\boldsymbol{s}_\Theta(\boldsymbol{x}_t^i, t)\|^2\right)\right| \geq \frac{\mathcal{R}}{2\hat{\kappa}^2}\sqrt{\frac{2t}{n}} =: t'\right)$$

$$+8\Pr\left(\sup_{\Theta \in \mathcal{B}_1}\left|\frac{1}{n}\sum_{i=1}^{n}2\zeta_i\left(\|\nabla \log q_t^s(\boldsymbol{x}_t^i)\|^2\right)\right| \geq \frac{\mathcal{R}}{2}\sqrt{\frac{2t}{n}} =: t''\right).$$

So, we need to find tail bounds for two concentration inequalities above: For the first one, we use our definition that $\|\Theta\|_1 \leq 1$ and $\sup_{\Theta \in \mathcal{B}_1}\|\boldsymbol{s}_\Theta(\boldsymbol{x}_t^i, t)\|_1 \leq 1$ that also implies $\sup_{\Theta \in \mathcal{B}_1}\|\boldsymbol{s}_\Theta(\boldsymbol{x}_t^i, t)\|^2 \leq 1$.

Following a similar uniform bound as proposed in Taheri et al. (2021, Proof of Theorem 5) for ReLU and regularized neural networks, we can obtain

$$\Pr\left(\sup_{\Theta\in\mathcal{B}_1}\left|\frac{1}{n}\sum_{i=1}^{n}\zeta_i\|s_\Theta(\boldsymbol{x}_t^i,t)\|^2\right|\geq\frac{\mathcal{R}}{4\hat{\kappa}^2}\sqrt{\frac{2t}{n}}=t'\right)\lesssim\frac{1}{n}$$

for $t=((\hat{\kappa}^2+(sB)^2)(2/L)^{2L}\log(n)\sqrt{L^2\log(p)\sum_{i=1}^{n}\|\boldsymbol{x}_i\|^4/n}/\mathcal{R})^2/2$, where $p$ stands for total number of network parameters and $L$ number of hidden layers of ReLU network. For simplicity, we can simplify the notation and shortly set $t=((\hat{\kappa}^2+(sB)^2)C_{\boldsymbol{x}}\sqrt{\log(pn)}/\mathcal{R})^2/2$, where $C_{\boldsymbol{x}}$ absorbs factors related to the input and all constants and we also used the fact that $(2/L)^{2L}L\leq 1$ for $L\geq 3$.

For the second concentration inequality, we use Hoeffding's inequality (Vershynin, 2018, Theorem 2.6.3) for $y_i=\zeta_i\|\nabla\log q_t^s(\boldsymbol{x}_t^i)\|^2$, that is $y_i=\zeta_i\|\nabla\log q_t^s(\boldsymbol{x}_t^i)\|^2$ are zero-mean random variables and bounded $\|\nabla\log q_t^s(\boldsymbol{x}_t^i)\|^2\leq s^2\|\nabla\log q_t^s(\boldsymbol{x}_t^i)\|_\infty^2\leq s^2B^2$, where we have used Assumption 4 and Assumption 3 to conclude that $\|\nabla\log q_t^s(\boldsymbol{x}_t^i)\|_\infty\approx\|\nabla\log q_t(\boldsymbol{x}_t^i)\|_\infty\leq B$. Now, we can progress as following:

$$\Pr\left(\sup_{\Theta\in\mathcal{B}_1}\left|\mathbb{E}_{X_t\sim Q_t}\|\hat{\kappa}s_\Theta(X_t,t)-\nabla\log q_t^s(X_t)\|^2-\frac{1}{n}\sum_{i=1}^{n}\|\hat{\kappa}s_\Theta(\boldsymbol{x}_t^i,t)-\nabla\log q_t^s(\boldsymbol{x}_t^i)\|^2\right|\geq 4\mathcal{R}\sqrt{\frac{2t}{n}}\right)$$

$$\leq 8\Pr\left(\sup_{\Theta\in\mathcal{B}_1}\left|\frac{1}{n}\sum_{i=1}^{n}\zeta_i\left(\|s_t(\boldsymbol{x}_t^i,t)\|^2\right)\right|\geq\frac{\mathcal{R}}{4\hat{\kappa}^2}\sqrt{\frac{2t}{n}}=t'\right)$$

$$+8\Pr\left(\left|\frac{1}{n}\sum_{i=1}^{n}\zeta_i\left(\|\nabla\log q_t^s(\boldsymbol{x}_t^i)\|^2\right)\right|\geq\frac{\mathcal{R}}{4}\sqrt{\frac{2t}{n}}=t''\right)$$

$$\leq\frac{16}{n}+16\exp\left(-\frac{nt''^2}{c's^4B^4}\right).$$

And using our assumptions we have

$$\mathcal{R}^2\leq\sup_{\Theta\in\mathcal{B}_1}\frac{1}{n}\sum_{i=1}^{n}\mathbb{E}_{X_t\sim Q_t}\|\hat{\kappa}s_\Theta(X_t^i,t)-\nabla\log q_t^s(X_t^i)\|^4$$

$$\leq 4\left(\hat{\kappa}^4+s^4B^4\right).$$

That leaves us with

$$\Pr\left(\sup_{\Theta\in\mathcal{B}_1}\left|\mathbb{E}_{X_t\sim Q_t}\|\hat{\kappa}s_\Theta(X_t,t)-\nabla\log q_t^s(X_t)\|^2-\frac{1}{n}\sum_{i=1}^{n}\|\hat{\kappa}s_\Theta(\boldsymbol{x}_t^i)-\nabla\log q_t^s(\boldsymbol{x}_t^i)\|^2\right|\right.$$

$$\left.\geq 8\sqrt{\hat{\kappa}^4+s^4B^4}\sqrt{\frac{2t}{n}}\right)$$

$$\leq\frac{16}{n}+16\exp\left(-\frac{nt''^2}{c's^4B^4}\right).$$

For $t=((\hat{\kappa}^2+(sB)^2)C_{\boldsymbol{x}}\sqrt{\log(pn)}/\mathcal{R})^2/2$, we then reach

$$\Pr\left(\sup_{\Theta\in\mathcal{B}_1}\left|\mathbb{E}_{X_t\sim Q_t}\|\hat{\kappa}s_\Theta(X_t,t)-\nabla\log q_t^s(X_t)\|^2-\frac{1}{n}\sum_{i=1}^{n}\|\hat{\kappa}s_\Theta(\boldsymbol{x}_t^i)-\nabla\log q_t^s(\boldsymbol{x}_t^i)\|^2\right|\right.$$

$$\left.\geq\left(\hat{\kappa}^2+s^2B^2\right)C_{\boldsymbol{x}}\sqrt{\frac{\log(pn)}{n}}\right)$$

$$\lesssim\frac{32}{n}.$$

$\square$

## B.5 PROOF OF LEMMA 3

*Proof.* The proof is based on some simple linear algebra and the definition of forward and backward processes.

We 1. use Bayes' rule, 2. consider a fixed $\boldsymbol{x}_t$ ($q_t(\boldsymbol{x}_t)$ is omitted since $\boldsymbol{x}_t$ is fixed), 3. definition of the forward process, and 4. multiplying with a one-valued factor and some rewriting, and 4. and some further rewriting

$$
\begin{aligned}
q_{t-1|t}(\boldsymbol{x}_{t-1}|\boldsymbol{x}_t) &= \frac{q_{t|t-1}(\boldsymbol{x}_t|\boldsymbol{x}_{t-1})q_{t-1}(\boldsymbol{x}_{t-1})}{q_t(\boldsymbol{x}_t)} \\
&\propto q_{t|t-1}(\boldsymbol{x}_t|\boldsymbol{x}_{t-1})q_{t-1}(\boldsymbol{x}_{t-1}) \\
&\propto q_{t-1}(\boldsymbol{x}_{t-1})\exp\left(-\frac{\|\boldsymbol{x}_t - \sqrt{\alpha_t}\boldsymbol{x}_{t-1}\|^2}{2(1-\alpha_t)}\right) \\
&\propto p_{t-1|t}^s(\boldsymbol{x}_{t-1}|\boldsymbol{x}_t)\exp\left(\log q_{t-1}(\boldsymbol{x}_{t-1}) + \frac{\alpha_t\|\boldsymbol{x}_{t-1} - \boldsymbol{u}_t^s\|^2}{2(1-\alpha_t)} - \frac{\|\boldsymbol{x}_t - \sqrt{\alpha_t}\boldsymbol{x}_{t-1}\|^2}{2(1-\alpha_t)}\right) \\
&= p_{t-1|t}^s(\boldsymbol{x}_{t-1}|\boldsymbol{x}_t)\exp\left(\log q_{t-1}(\boldsymbol{x}_{t-1}) + \frac{\alpha_t\|\boldsymbol{x}_{t-1} - \boldsymbol{u}_t^s\|^2}{2(1-\alpha_t)} - \frac{\alpha_t\|\boldsymbol{x}_{t-1} - (\boldsymbol{x}_t/\sqrt{\alpha_t})\|^2}{2(1-\alpha_t)}\right)
\end{aligned}
$$

using the fact that (see equation 7)

$$
p_{t-1|t}^s(\boldsymbol{x}_{t-1}|\boldsymbol{x}_t) \propto \exp\left(-\frac{\alpha_t\|\boldsymbol{x}_{t-1} - \boldsymbol{u}_t^s\|^2}{2(1-\alpha_t)}\right).
$$

We then use the fact that

$$
\begin{aligned}
\|\boldsymbol{x}_{t-1} - \boldsymbol{u}_t^s\|^2 - \|\boldsymbol{x}_{t-1} - (\boldsymbol{x}_t/\sqrt{\alpha_t})\|^2 &= \|\boldsymbol{x}_{t-1} - (\boldsymbol{x}_t/\sqrt{\alpha_t}) + (\boldsymbol{x}_t/\sqrt{\alpha_t}) - \boldsymbol{u}_t^s\|^2 - \|\boldsymbol{x}_{t-1} - (\boldsymbol{x}_t/\sqrt{\alpha_t})\|^2 \\
&= 2\big(\boldsymbol{x}_{t-1} - (\boldsymbol{x}_t/\sqrt{\alpha_t})\big)^T\big((\boldsymbol{x}_t/\sqrt{\alpha_t}) - \boldsymbol{u}_t^s\big) + \|(\boldsymbol{x}_t/\sqrt{\alpha_t}) - \boldsymbol{u}_t^s\|^2.
\end{aligned}
$$

We then use equation 7 to obtain

$$
\begin{aligned}
\frac{2\big(\boldsymbol{x}_{t-1} - (\boldsymbol{x}_t/\sqrt{\alpha_t})\big)^T\big((\boldsymbol{x}_t/\sqrt{\alpha_t}) - \boldsymbol{u}_t^s\big)}{(1-\alpha_t)/\alpha_t} &= -\big(\boldsymbol{x}_{t-1} - (\boldsymbol{x}_t/\sqrt{\alpha_t})\big)^T\sqrt{\alpha_t}\nabla\log q_t^s(\boldsymbol{x}_t) \\
&= -\sqrt{\alpha_t}\boldsymbol{x}_{t-1}^T\nabla\log q_t^s(\boldsymbol{x}_t) + \boldsymbol{x}_t^T\nabla\log q_t^s(\boldsymbol{x}_t).
\end{aligned}
$$

Collecting all the pieces above we obtain for a fixed $\boldsymbol{x}_t$

$$
q_{t-1|t}(\boldsymbol{x}_{t-1}|\boldsymbol{x}_t) \propto p_{t-1|t}^s(\boldsymbol{x}_{t-1}|\boldsymbol{x}_t)\exp\big(\zeta_{t,t-1}(\boldsymbol{x}_t, \boldsymbol{x}_{t-1})\big)
$$

with $\zeta_{t,t-1}(\boldsymbol{x}_t, \boldsymbol{x}_{t-1}) = \log q_{t-1}(\boldsymbol{x}_{t-1}) - \sqrt{\alpha_t}\boldsymbol{x}_{t-1}^T\nabla\log q_t^s(\boldsymbol{x}_t) + f(\boldsymbol{x}_t)$, where $f(\boldsymbol{x}_t)$ can be considered as an arbitrary function of $\boldsymbol{x}_t$, since $\boldsymbol{x}_t$ was fixed (let's also note that the term $\|(\boldsymbol{x}_t/\sqrt{\alpha_t}) - \boldsymbol{u}_t^s\|^2$ is omitted since it is just dependent over $\boldsymbol{x}_t$). That completes the proof.

$\square$

### B.6 PROOF OF LEMMA 4

*Proof.* We employ the fact that $\hat{p}$ and $p^s$ are both Gaussian with the same variance, $\widehat{P}_{t-1|t} = \mathcal{N}(\boldsymbol{x}_{t-1}; \hat{\boldsymbol{u}}_t(\boldsymbol{x}_t), \sigma_t^2\mathbb{I}_d)$ and $P_{t-1|t}^s = \mathcal{N}(\boldsymbol{x}_{t-1}; \boldsymbol{u}_t^s(\boldsymbol{x}_t), \sigma_t^2\mathbb{I}_d)$ with

$$
\hat{\boldsymbol{u}}_t(\boldsymbol{x}_t) = \frac{1}{\sqrt{\alpha_t}}\big(\boldsymbol{x}_t + (1-\alpha_t)\hat{\kappa}\boldsymbol{s}_{\widehat{\Theta}_{\ell_1}}(\boldsymbol{x}_t, t)\big)
$$

and

$$
\boldsymbol{u}_t^s(\boldsymbol{x}_t) = \frac{1}{\sqrt{\alpha_t}}\big(\boldsymbol{x}_t + (1-\alpha_t)\nabla_{\boldsymbol{x}_t}\log q_t^s(\boldsymbol{x}_t)\big).
$$

We use 1. the Gaussian property, 2. rewriting, 3. add a zero-valued term, 4. use the property that $\mathbb{E}_{X_{t-1}\sim\mathbb{Q}_{t-1|t}}[X_{t-1}|\boldsymbol{x}_t] = \boldsymbol{u}_t(\boldsymbol{x}_t)$, 5. definitions of $\boldsymbol{u}_t$, $\boldsymbol{u}^s$, and $\hat{\boldsymbol{u}}_t$ and Cauchy–Schwarz inequality

to obtain

$$\mathbb{E}_{X_t, X_{t-1} \sim \mathbb{Q}_{t,t-1}} \left[ \log \frac{p_{t-1|t}^s(X_{t-1}|X_t)}{\hat{p}_{t-1|t}(X_{t-1}|X_t)} \right]$$

$$= \mathbb{E}_{X_t, X_{t-1} \sim \mathbb{Q}_{t,t-1}} \left[ \frac{\alpha_t}{2(1-\alpha_t)} (\|X_{t-1} - \hat{\boldsymbol{u}}_t(X_t)\|^2 - \|X_{t-1} - \boldsymbol{u}^s(X_t)\|^2) \right]$$

$$= \mathbb{E}_{X_t, X_{t-1} \sim \mathbb{Q}_{t,t-1}} \left[ \frac{\alpha_t}{(1-\alpha_t)} \left(X_{t-1} - \boldsymbol{u}_t^s(X_t)\right)^T \left(\boldsymbol{u}_t^s(X_t) - \hat{\boldsymbol{u}}_t(X_t)\right) \right.$$

$$\left. + \frac{\alpha_t}{2(1-\alpha_t)} \|\boldsymbol{u}_t^s(X_t) - \hat{\boldsymbol{u}}_t(X_t)\|^2 \right]$$

$$= \mathbb{E}_{X_t, X_{t-1} \sim \mathbb{Q}_{t,t-1}} \left[ \frac{\alpha_t}{(1-\alpha_t)} \left(X_{t-1} - \boldsymbol{u}_t(X_t) + \boldsymbol{u}_t(X_t) - \boldsymbol{u}_t^s(X_t)\right)^T \left(\boldsymbol{u}_t^s(X_t) - \hat{\boldsymbol{u}}_t(X_t)\right) \right.$$

$$\left. + \frac{\alpha_t}{2(1-\alpha_t)} \|\boldsymbol{u}_t^s(X_t) - \hat{\boldsymbol{u}}_t(X_t)\|^2 \right]$$

$$= \mathbb{E}_{X_t, X_{t-1} \sim \mathbb{Q}_{t,t-1}} \left[ \frac{\alpha_t}{(1-\alpha_t)} \left(\boldsymbol{u}_t(X_t) - \boldsymbol{u}_t^s(X_t)\right)^T \left(\boldsymbol{u}_t^s(X_t) - \hat{\boldsymbol{u}}_t(X_t)\right) \right.$$

$$\left. + \frac{\alpha_t}{2(1-\alpha_t)} \|\boldsymbol{u}_t^s(X_t) - \hat{\boldsymbol{u}}_t(X_t)\|^2 \right]$$

$$\leq (1-\alpha_t) \sqrt{\mathbb{E}_{X_t \sim Q_t} \|\nabla \log q_t(X_t) - \nabla \log q_t^s(X_t)\|^2 \mathbb{E}_{X_t \sim Q_t} \|\nabla \log q_t^s(X_t) - \hat{\kappa} \boldsymbol{s}_{\widehat{\Theta}_{\ell_1}}(X_t, t)\|^2}$$

$$+ \frac{1-\alpha_t}{2} \mathbb{E}_{X_t \sim Q_t} \|\hat{\kappa} \boldsymbol{s}_{\widehat{\Theta}_{\ell_1}}(X_t, t) - \nabla \log q_t^s(X_t)\|^2.$$

That implies

$$\sum_{t=1}^T \mathbb{E}_{X_t, X_{t-1} \sim \mathbb{Q}_{t,t-1}} \left[ \log \frac{p_{t-1|t}^s(X_{t-1}|X_t)}{\hat{p}_{t-1|t}(X_{t-1}|X_t)} \right]$$

$$\leq \sum_{t=1}^T (1-\alpha_t) \sqrt{\mathbb{E}_{X_t \sim Q_t} \|\nabla \log q_t(X_t) - \nabla \log q_t^s(X_t)\|^2 \mathbb{E}_{X_t \sim Q_t} \|\nabla \log q_t^s(X_t) - \hat{\kappa} \boldsymbol{s}_{\widehat{\Theta}_{\ell_1}}(X_t, t)\|^2}$$

$$+ \sum_{t=1}^T \frac{1-\alpha_t}{2} \mathbb{E}_{X_t \sim Q_t} \|\hat{\kappa} \boldsymbol{s}_{\widehat{\Theta}_{\ell_1}}(X_t, t) - \nabla \log q_t^s(X_t)\|^2,$$

as desired.

□

