# OpenReview forum: "Regularization can make diffusion models more efficient"
_ICLR.cc/2026/Conference — Submitted to ICLR 2026_

### Official Review · Reviewer_5MyW · 2025-10-20

**Soundness:** 3
**Presentation:** 3
**Contribution:** 3
**Rating:** 6
**Confidence:** 3

**Summary:**

This paper proposes a regularization technique to accelerate diffusion models, with the core concept being the regularized denoising score-matching estimator (Equation 3). Based on this framework, the training efficiency of diffusion models is improved. The authors demonstrate the effectiveness of the method through both theoretical analysis and experimental validation.

**Strengths:**

A key strength of this paper lies in its theoretical rigor. The mathematical derivations of Theorem 1 are solid and clearly articulate the core conclusion. This high standard of rigor and clarity is also reflected in the mathematical sections and the explanation of the methods and motivations. Finally, the experimental results effectively validate this theoretical framework.

**Weaknesses:**

**Major concerns**
+ The TOY EXAMPLE in Figure 1 is confusing. The author states that “regularized denoising score matching predominantly adheres to the two-dimensional sub-manifold (along the Y and Z axes).” Can samples on both sides of the X-axis be effectively generated? Would this lead to a decrease in generated diversity?

+ The paper's evaluation currently relies primarily on visual comparisons, which are insufficient on their own. As previously mentioned, the manuscript does not include common quantitative metrics for generative models, such as Precision and Recall. The authors should incorporate these (or other relevant) metrics to provide a more objective assessment of the method's performance, specifically regarding its ability to approximate the original data (fidelity, Precision) and generate diverse samples (diversity, Recall).

> Improved Precision and Recall Metric for Assessing Generative Models. NeurIPS-2019

+ The current diffusion models adopt the Flow Matching framework, which constructs different intermediate distributions and shifts the model's prediction target from a score function to a velocity field. Can the conclusions of this paper be generalized to this setting?

> Flow Matching for Generative Modeling. arXiv:2210.02747

**Minor concerns**
+ The presentation of the figures could be improved. For instance, in Figures 2 and 3, enhancing the visual contrast between GT, baseline, and the proposed method would make the comparisons clearer and more impactful.

**Questions:**

+ Regarding the practical implementation of equation 3, **it is unclear how the parameters are constrained to satisfy the requirements of equation  2**. Clarification on this mechanism is needed. If the author could provide pseudocode for the implementation, it would make the article more persuasive.

+ Figure 5 shows relatively high FID values for CIFAR, while most diffusion models currently achieve FID values below 5. Could the authors explore more advanced network architectures to validate the method's generalizability?

---

> ### Author Response · Authors · 2025-11-20
> **Response**
>
> We thank the reviewer for carefully reading our paper and for their positive evaluation. We greatly appreciate the time taken to assess our work. We would carefully respond to the raised questions.
>
> **TOY EXAMPLE in Figure 1** : Thanks for raising this point. As sample generation starts randomly from a point drawn from the Gaussian distribution, in practice, all directions of the true distribution can be equally explored by starting from different random initial points. To support our claim, we evaluated the quality of the 2000 generated 3D samples by comparing them to the true distribution using several metrics: **Fréchet distance**, **MMD²** with an RBF kernel, **Energy distance**, as well as error mean and covariances. Across all these measures, the samples obtained with the regularized score-matching objective consistently show lower distances to the ground truth compared to the original score-matching setup. This demonstrates that our regularized approach clearly outperforms the original method in capturing not only the mean and covariance but also the higher-order structure of the target distribution.
>
> Mean error:        org=0.27   reg=0.26
>
> Cov error (Frob):  org=1.12    reg=1.09
>
> Frechet distance:  org=0.75         reg=0.53
>
> MMD^2 (RBF):      org=0.12       reg=0.03
>
> Energy distance:   org=0.30         reg=0.19
>
> **New simulations with large model size for CIFAR with FID metrics**
> We now conduct the CIFAR-10 experiments using the standard model size (base channel 128). The original score-matching setup achieves an $\operatorname{FID}$ of approximately $25$, consistent with the results reported in Song et al., 2019. Using our regularized objective with $r=10^{-4}$, we further improve the $\operatorname{FID}$ to $23$.
> Under a small time budget of $T=200$, the performance of the original score-matching method deteriorates significantly, yielding an $\operatorname{FID}$ of $160$, whereas our approach achieves an $\operatorname{FID}$ of $49$ with $r=5 \times 10^{-4}$. This demonstrates that our method consistently outperforms standard score matching, both under tight time budgets and for large $T$. We will include the corresponding results and generated samples in the main PDF.
>
> **Flow Matching framework**: Thanks for raising this question. The reviewer is correct—our approach can generally be applied from a flow matching perspective, learning sparse vector fields. In fact, diffusion models can be viewed as a special case of flow matching, making this connection natural and relevant.
>
> **Practical implementation of equation 3**: Good question. We have indeed implemented the $\ell_1$-norm constraints as additional regularizers in the objective function. We will make sure to provide pseudocode in the updated version.
>
> **Improving FID values for CIFAR**: We have now implemented a larger network to improve FID scores, aligning with the results reported in Song et al. (2019) (Please see above). Achieving even lower FID values would likely require the use of more advanced samplers, which represents a new research direction. In this work, we focus on the basic sampler in order to isolate and study the impact of regularization.
>
> We thank the reviewer once again for their input. We will incorporate the above-mentioned comments and update the presentation of the figures accordingly. Please let us know if anything remains unclear.

---

> > ### Comment · Reviewer_5MyW · 2025-11-27
> >
> > Thank you for the response. My confusion has been well resolved, and I will maintain the positive rating.

---

### Official Review · Reviewer_La1N · 2025-11-01

**Soundness:** 4
**Presentation:** 4
**Contribution:** 4
**Rating:** 8
**Confidence:** 3

**Summary:**

This paper investigates how introducing l1-based regularization into score matching can make diffusion models more efficient, both theoretically and empirically. The authors assume that the true score function of the data distribution is approximately s-sparse, meaning that only a few dimensions matter for denoising. Under this sparsity assumption, they derive non-asymptotic bounds showing that the KL divergence scales with $s^2$ rather than $d^2$, and validate this through experiments on toy data and image datasets, where the regularized model achieves comparable or better quality with much fewer sampling steps.

**Strengths:**

1. Provides a clear theoretical link between sparsity regularization and improved non-asymptotic convergence, reducing dimensional dependence from $d$ to $s$.
2. The proposed regularized score-matching objective is simple to implement and compatible with existing diffusion frameworks

**Weaknesses:**

The main theoretical guarantee hinges on the unverified assumption that the true score function is s-sparse, and given the experiments are limited to low-dimensional datasets, it remains unclear whether the same efficiency extends to large-scale diffusion models.

**Questions:**

See Weakness

---

> ### Author Response · Authors · 2025-11-20
>
> We sincerely thank the reviewer for the thorough and positive assessment of our work. We greatly appreciate the constructive feedback and comments.  We would now clarify the point raised by the reviewer about our sparsity assumption.
>
> **Sparsity assumption**: We would like to draw your attention to our detailed discussion of the sparsity assumption in Section ''DISCUSSION OVER ASSUMPTION 4'' in the Appendix. There, we specifically show that in the worst-case scenario, where the true score function is not sparse, our regularized estimator performs comparably to the original score-matching method, demonstrating that it either matches or improves upon the original estimator. We then provide a discussion when the sparsity assumption holds (for datasets with features carrying near-uniform distribution). We also provide numerical evidence verifying sparsity on the Toy and MNIST datasets. Finally, we relax Assumption 4 to the more general Assumption 5, broadening the applicability of our results.
>
> We thank the reviewer once again for the positive evaluation and would be happy to address any remaining questions.

---

> > ### Comment · Reviewer_La1N · 2025-11-27
> >
> > The authors’ response has addressed my earlier concerns. I will keep my original score of 8

---

### Official Review · Reviewer_fKXu · 2025-11-03

**Soundness:** 2
**Presentation:** 2
**Contribution:** 2
**Rating:** 2
**Confidence:** 4

**Summary:**

This paper proposes to accelerate score-based diffusion model inference by introducing sparsity regularization into the score function. Theoretically, the authors provide proofs supporting the feasibility of the approach; however, the experimental results do not sufficiently validate these claims. The paper would benefit from an expanded experimental section, removal of the “paper outline” section, and more compact figure layouts to better utilize space and improve readability.

**Strengths:**

The paper provides detailed explanations for each assumption and theorem, offering clear intuition behind the proposed method. The theoretical derivations are comprehensive and rigorous, establishing a solid foundation for the concept of regularized score-based diffusion models.

**Weaknesses:**

1. The paper does not provide code or implementation details, making it difficult to assess the practical feasibility of the method. Moreover, there is no analysis of the hyperparameters $s,r,\kappa$, leaving the experimental section incomplete.

2. From the presented results, the method only shows effectiveness on datasets with inherently sparse structures, such as MNIST. Even for MNIST, where sparsity is visually evident, the sparsity-inducing SGM struggles to generate high-quality samples with a small number of timesteps. In Figure 2 (right column), the generated images at T=50 and T=20 still contain noticeable noise, and this issue becomes even more pronounced in Figures 3, 4, and 5.

3. Minor comment: The description of the diffusion process (Lines 29–34) should use consistent notation for the variable $x$ to avoid confusion.

**Questions:**

1. The sparsity-inducing regularization requires solving the optimization problem in Eq. (3), which introduces additional computational overhead. Although the paper emphasizes inference-time acceleration, it is important to analyze how this optimization affects training time and training stability.

2. Sparsity-inducing methods are typically applied for one of two purposes: (a) when genuine sparsity exists in the data or model structure with interpretable meaning, or (b) to improve computational efficiency in systems involving heavy matrix operations. However, diffusion processes do not inherently involve such high-order matrix computations. The authors should better justify the motivation. Why does enforcing sparsity in the score function improve inference efficiency, and under what conditions is this beneficial?

3. Adding sparsity constraints might reduce the model’s generalization ability to fit data distributions, especially for high-dimensional modalities such as images or text. How does the proposed method balance sparsity with the ability to capture complex data distributions?

---

> ### Author Response · Authors · 2025-11-20
>
> We thank the reviewer for their input and for finding our theoretical derivations comprehensive and rigorous. We greatly appreciate the time taken to assess our work. We would carefully respond to the raised questions.
>
> **Training time and training stability** : We would like to draw the reviewer’s attention to the fact that $\ell_1$-regularized estimators are standard in machine learning and statistics (Lederer, 2022b; Eldar & Kutyniok, 2012). Although the $\ell_1$-norm is non-smooth, it typically poses very few practical challenges in terms of computation (see Friedman et al., 2010). To support this claim, we measured the optimization time for training both the regularized and the original diffusion models on MNIST for $50$ epochs using Adam on the same device. The training time for the original DFM is approximately $20$ minutes, while the regularized version requires about $21$ minutes, and both objectives converge after roughly $20$ epochs. This demonstrates that adding $\ell_1$ regularization has only a minor impact on training time. Similar behavior was observed for other datasets. We will incorporate these results into the final version.
>
> **Motivation for employiong sparsity** :  We highlight that our regularization technique is motivated by two main considerations:
> First, it allows for a more focused search during sample generation by concentrating on the features
> that have the greatest influence on the generated samples (see Figure 1, right panel).
> Second, as noted in [Ren2025, Example 4.1], one of the dominant sources of error in the convergence of diffusion models
> is the estimation error
> $$
> E_{{est}}(\Theta)
>  = E_{x_0\sim q_0} \int_0^T
>     E_{x_t\sim q_{t|0}}
>       ||s_{\Theta}(x_t,t)-\nabla\log q_{t|0}(x_t|x_0)||^2
>      dt ,
> $$
> which, in the finite-sample regime, can lead to overfitting.
> Thus, regularization—being a classical and effective remedy for overfitting—naturally plays a crucial role in reducing estimation error. Please also see our Corollary 1, which provides sample complexity.
>
> [1] Y. Ren, G. Rotskoff, and L. Ying. A unified approach to analysis and design of denoising markov models. arXiv:2504.01938, 2025.
>
> **Sparsity constraints might reduce the model’s generalization**:  Let us first emphasize that our regularization technique is motivated by the work of Taheri et al. (2021), which showed that scale regularization can recover the true radius of the underlying target. In a similar spirit, our approach identifies an appropriate function space to explore and therefore does not suffer from approximation loss. We would also like to draw your attention to our detailed discussion of the sparsity assumption in Section “DISCUSSION OVER ASSUMPTION 4” of the Appendix. There, we show that in the worst-case scenario—namely, when the true score function is not sparse—our regularized estimator performs on par with the original score-matching method, ensuring that it never performs worse and often performs better.
>
> **Effectiveness on datasets with inherently sparse structures**: Please note that we presented figures under an extremely tight time budget (1/20  smaller time budget), under which the original score-matching method completely fails to generate meaningful samples, while our approach still succeeds. This choice was intentional to clearly highlight the differences. Under moderate time budgets, our method produces samples that are qualitatively comparable to those obtained with a high budget. We will include the corresponding images in the updated version. For example, for CIFAR10, in our new simulations, by moving from $T=1000$ to $T=500$, our FID drops just 2 units. While the original score matching shows a major drop ($13$ units).
>
> **Analysis of the hyperparameters**: We would like to draw your attention to our simulations under  DISCUSSION OVER ASSUMPTION 4  Section in the appendix, where we study the sparsity assumption for Toy and MNIST data.
>
> For tuning $r$,   as already mentioned in the paper, the tuning parameters are chosen as a decreasing function of T, say $r = f[T] \in O(c'/T)$ for some constant $c' \in (0,\infty)$. In almost all of our experiments, we used a tuning of order $1\mathrm{e}{-4}$  for large T and $1\mathrm{e}{-3}$ for small T (for CIFAR10, we used tuning $5e-4$ for small $T$ as the level of sparsity is lower than the MNIST family). Our empirical observations indicate that small changes in this order have only a minor impact on the results (results are stable for  $r \in [1e-4,5e-4)$). Moreover, the tuning needs to be selected only during the training phase---as diffusion models involve several hyperparameters to be set---while once the model is trained, it can be used for inference without further adjustment.
>
>
> **Code**: We will make sure to upload our code by the end of the discussion time. Thanks.
>
> We will also make sure to implement the above-mentioned changes to our new version; please let us know if any questions remain open.

---

> > ### Comment · Reviewer_fKXu · 2025-11-21
> >
> > Thank you for the authors’ response to my questions. I hope you will further improve the paper accordingly. I know that $l_1$ regularization is a classical optimization problem with well-established solvers; my concern, however, is whether introducing $l_1$ optimization may slow down the convergence of the diffusion loss. The authors mentioned that experiments show comparable convergence speeds, which indeed addresses my concern. I strongly recommend adding this supplementary experiment to the paper.
> >
> > Regarding Assumption 4 on toy data and MNIST in the appendix, I suggest moving this content to the main text and providing more detailed explanations. In the main text, the sparsity level is described as $s \in $ {1, 2, …}, whereas the appendix reports sparsity levels of {600, 400, 200, 100, 20}, which seem to be on a very different scale.
> >
> > More importantly, the purpose of introducing the $l_1$ constraint is usually to argue that the feature space of the data is indeed sparse. The datasets used in the experiments (e.g., toy data and MNIST) also exhibit clear sparse structures. However, more complex real-world datasets such as ImageNet possess significantly more complicated feature spaces. I remain unconvinced about the practical utility of $l_1$ sparsity in such scenarios, especially since the CIFAR-10 results already show minimal difference between using and not using the $l_1$ constraint.
> >
> > If the primary motivation is simply to obtain a more focused search during sample generation, then why not directly adopt classifier-free guidance (CFG), a much simpler and more established technique for enhancing generation quality? Or if it is to avoid overfitting, why do you choose $l_1$ norm rather than other norm (e.g. $l_2$ norm)? In addition, since the sparsity-induced norm is applied directly to the score, could the authors visualize and compare the sparse and unsparse scores for MNIST and CIFAR-10? Such a comparison would clearly demonstrate whether the $l_1$ constraint is indeed necessary.
> >
> > While I am willing to increase my score to 4, I still believe that the paper is currently below the acceptance threshold.

---

> ### Author Response · Authors · 2025-11-21
>
> We thank the reviewer for their prompt response and are glad that we could address your questions; we will make sure to imply proposed changed. We will clarify the newly raised questions below:
>
> **Practical utility of sparsity for CIFAR-10** : We would draw the reviewers' attention to our CIFAR-10 simulation below, which clearly shows the power of $\ell_1$-regularization not only in a tight time budget but also for moderate and large T:
>
> **New simulations with large model size for CIFAR with FID metrics:**
> We conducted a new simulation on CIFAR-10 using the standard model size (base channel 128). The original score-matching setup achieves an $\operatorname{FID}$ of approximately $25$, consistent with the results reported in Song et al., 2019. Using our regularized objective with $r=10^{-4}$, we further improve the $\operatorname{FID}$ to $23$.
> For $T=500$, our regularized approach ends with $FID=25$, while original score matching drops to $38$.
> Under a small time budget of $T=200$, the performance of the original score-matching method deteriorates significantly, yielding an $\operatorname{FID}$ of $160$, whereas our approach achieves an $\operatorname{FID}$ of $49$ with $r=5 \times 10^{-4}$. This demonstrates that our method consistently outperforms standard score matching, both under tight time budgets and for large $T$. We will include the corresponding results and generated samples in the main PDF.
>
> **Classifier-free guidance** :  While classifier-free guidance (CFG) is indeed a powerful technique for **conditional diffusion models**, it serves a different purpose from our method. To the best of our knowledge, CFG functions as an inference-time sampler that steers the generation process without modifying the underlying score network. In contrast, our approach directly targets the structural properties and training dynamics of the model itself, and  CFG is actually known as a predictor-corrector sampler (Bradley & Nakkiran (2024)).
> Furthermore, CFG is designed specifically for **conditional settings** and has documented limitations. For instance, Shen et al. (2024) demonstrate that using a single global CFG scale can introduce spatial inconsistencies across regions with varying semantic strengths, resulting in suboptimal image quality. So, one can view CFG as a complementary technique rather than a replacement for our proposed approach.
>
> **$\ell_2$-regularization**: Although $\ell_2$-norm helps control the growth of magnitudes, it does not promote sparsity by design (see  Lederer2022), and so, can not stop overfitting.
> Our complementary simulations show that using $\ell_2$-regularization does not significantly speed up inference time in our simulations (please see the COMPARISON WITH OTHER SPARSITY-INDUCING REGULARIZERS section in the Appendix).
>
> J. Lederer. Fundamentals of High-Dimensional Statistics: with exercises and R labs. Springer Texts
> in Statistics, 2022.
>
> **Scale of sparsity**: True; let us refer to our discussion under Assumption 4 in the text:
> ''Naturally, the sparsity level $s$  may vary depending on the dataset and could grow at different rates in different contexts. We refer the reader to the detailed discussion following Theorem 1 and in Section A.3, where we analyze the worst-case scenario for the scaling behavior of ϵ relative to the rate of the tuning parameter r. Also, we refer to Assumption 5 in the supplementary material for a more relaxed version of our sparsity assumption, which can be used in place of Assumption 4.'' We show in Section A.3 that in the worst-case scenario—namely, when the true score function is not sparse—our regularized estimator performs on par with the original score-matching method, ensuring that it never performs worse and often performs better. Our simulations clearly show this claim (see CIFAR-10 metrics above). Please also note that, since no ground-truth scores are available for MNIST, the computed errors (over different sparsity levels) should be considered pessimistic estimates, as they are affected by the estimation error of original score matching.
>
> **Since the sparsity-induced norm is applied directly to the score, could the authors visualize and compare the sparse and unsparse scores for MNIST and CIFAR-10?** Could the reviewer please clarify the proposed simulation so that we can design it appropriately? For example, would you like us to compare the level of sparsity in the estimated scores under our method versus the original DFM for MNIST and CIFAR10 (perhaps by plotting a histogram)?
>
> We thank the reviewer once again for the discussion.

---

> ### Author Response · Authors · 2025-11-24
>
> Following the question raised by the reviewer: **Since the sparsity-induced norm is applied directly to the score, could the authors visualize and compare the sparse and unsparse scores for MNIST and CIFAR-10?**,  we have conducted a new experiment:
>  In this experiment, we compare the score function produced by the original model with that of the regularized model across the full diffusion time horizon t∈[0,1]. For a fixed batch of inputs, we evaluate both models at many noise levels t  and measure two quantities:
>
> (1) L0 sparsity (fraction of score entries exceeding a small threshold) and
>
> (2) the L1 magnitude of the score.
>
>
> By averaging these quantities over all time t (and also separately analyzing for low-noise regions), we quantify how much the regularization reduces the score magnitude and induces sparsity throughout the diffusion process.
>
> For **CIFAR-10**, the results show a $12$% reduction in the
> $\ell_{0}$-activity (fraction of non-negligible score entries) for the
> regularized model when averaged over all time steps, and an $17$% reduction
> in the low-noise region $(0.0, 0.2]$.
> The average $\ell_{1}$-norm of the score decreases from $1.15$ (original) to $0.90$ (regularized) over
> the full time horizon, and from $3.15$ to $2.40$ in the low-noise region.
>
> For **MNIST**, the results show a $15$% reduction in the
> $\ell_{0}$-activity for the
> regularized model when averaged over all time steps, and a $19$% reduction
> in the low-noise region $(0.0, 0.2]$.
> The average $\ell_{1}$-norm of the score decreases from $0.93$ (original) to $0.63$ (regularized) over
> the full time horizon, and from $3.06$ to $1.66$ in the low-noise region.
>
> These results indicate that sparsity is more pronounced in low-noise regions, where the data maintains its inherent structure.
>
> We hope this addresses the reviewer’s question. Please let us know if any further clarification is needed.
> We will provide these results in our updated version.

---

> ### Author Response · Authors · 2025-11-26
> **Response**
>
> As raised by the reviewer, we also provide the source code. Due to file size limitations, we were unable to include the model checkpoints.

---

> > ### Comment · Reviewer_fKXu · 2025-11-28
> >
> > Thank you for providing the additional Python notebook. This improves the credibility and reproducibility of the experimental results. For the CIFAR-10 experiments, the FID comparisons indeed show that introducing the sparsity-inducing norm has observable effects on more complex datasets. Your clarifications regarding the L2-normalization and the scale of sparsity are also helpful.
> >
> > However, some of my concerns remain unresolved.
> >
> > - First, I believe you should include classifier-free guidance (CFG) as a benchmark. Although your method is technically different, the motivation appears similar: both aim to achieve a more focused search during sample generation (see Fig. 1 of [Guiding a Diffusion Model with a Bad Version of Itself](https://arxiv.org/pdf/2406.02507) and Fig. 1 of your paper). It would be useful to compare how different time steps influence FID under both approaches. Because the experimental section of the paper is relatively weak overall, and there is no comparison with the benchmark, I think this is a feasible way to enrich the experimental content.
> >
> > - Second, regarding the experiment on comparing sparse versus unsparse scores for MNIST and CIFAR-10: the score function is fundamentally a gradient vector field in terms of images. Since the sparsity-inducing norm is applied directly to the score, I do not yet see a clear motivation for why the score should exhibit inherent sparsity. This is why I have repeatedly asked about the underlying motivation. Without evidence that the score field truly possesses a sparse structure, applying a sparsity regularizer appears less theoretically grounded.
> >
> > If you could visualize and compare the sparse and unsparse score fields, demonstrate that the score indeed exhibits sparsity, and incorporate these results into the paper, I would be able to raise my score to a 6. Otherwise, as previously discussed, I will need to maintain my current rating of 4.

---

> ### Author Response · Authors · 2025-12-01
>
> We thank the reviewer for their input. We have now incorporated the proposed simulation into the revised version. Specifically, we added a new section, **Sparsity within Scores,** in the Appendix, where we examine the sparsity properties of score functions for MNIST and CIFAR-10 images. This section includes heat maps of the scores (for both original and regularized DFM) as well as box plots that illustrate sparsity patterns (Figures 7 and 8), clearly supporting our sparsity motivation and perspective.
>
> Regarding classifier-free guidance (CFG), we fully agree that it is a well-known and widely used technique for accelerating diffusion sampling, and we will make sure to explicitly acknowledge this in the revised manuscript. However, it is important to emphasize once more that CFG is fundamentally different in nature from our approach: CFG is a **conditional guidance mechanism** (typically relying on prompts, conditioning models, ...), whereas our method operates in a purely unconditional setting.

---

### Official Review · Reviewer_Hjd6 · 2025-11-03

**Soundness:** 3
**Presentation:** 3
**Contribution:** 2
**Rating:** 4
**Confidence:** 4

**Summary:**

This paper proposes adding a type of L^1 regularization to the score-matching objective in diffusion, and analyzes the iteration complexity of sampling from this model, and compares it to the standard score-matching objective -- it shows that convergence can be achieved using a number of iterations depending on the sparsity of the score function, rather than its ambient dimension. The paper shows empirical results on a synthetic task, MNIST and FashionMNIST to support its theoretical claims, and to support the assumption that score functions are sparse in practice.

**Strengths:**

The addition of an explicit L^1 regularization to the score-matching objective, and a guarantee on sampling iteration complexity depending on the sparsity of the score functions is interesting -- it's further interesting that the new objective does seem to give reasonable results on (very) small-scale experiments. The fact that the new objective provides a reasonable approximation to the score even in the absence of sparsity is a plus. The paper itself is easy to read, and the contributions are laid out clearly.

Overall, I like the contribution of this paper at a conceptual level, and the small-scale/initial empirical results are intriguing.

**Weaknesses:**

From a technical perspective, the theory seems to largely be a rehashing of previous results with some modifications. The empirical results are unfortunately too small-scale to be convincing to practitioners -- even the CIFAR experiments included in the appendix were run using a non-standard 32 channel network instead of the standard 128 channels.

Additionally, it's not clear how the regularization parameter $r$ should be set in practice, and for instance, how sensitive the results are to this setting. The added complexity in the training could be a huge drawback for practitioners.

The theory could be made stronger by analyzing the *sample complexity* of learning the score using the new objective under sparsity constraints -- see [1] and [2]. I'd also be interested in knowing whether the recent more sophisticated sampling methods and analyses (see [3], [4]) translate to the sparse case analyzed in this paper.

[1]: Generative modeling with denoising auto-encoders and langevin sampling. A Block, Y Mroueh, A Rakhlin. 2020

[2]: Improved Sample Complexity Bounds for Diffusion Model Training. Shivam Gupta, Aditya Parulekar, Eric Price, and Zhiyang Xun. NeurIPS 2024.

[3]: Faster Diffusion Sampling with Randomized Midpoints: Sequential and Parallel. Shivam Gupta, Linda Cai, Sitan Chen. ICLR 2025.

[4]: Nearly -Linear Convergence Bounds for Diffusion Models via Stochastic Localization. J Benton, V De Bortoli, A Doucet, G Deligiannidis. ICLR 2024.

**Questions:**

1) What is the sample complexity of learning sparse scores using the objective you have proposed?
2) Can you achieve iteration complexity smaller than s^2 using the recent more sophisticated sampling algorithms and analyses?
3) Do the results translate to larger scale? For example LSUN or Celeb-A?
4) There are no quantitative results provided for many of the small scale experiments (MNIST, etc) -- can you report them?

---

> ### Author Response · Authors · 2025-11-20
> **Response**
>
> We thank the reviewer for carefully reading our paper and for their positive evaluation. We greatly appreciate the time taken to assess our work. We would carefully respond to the raised questions.
>
> **New simulations with large model size for CIFAR with FID metrics**
> We now conduct the CIFAR-10 experiments using the standard model size (base channel 128). The original score-matching setup achieves an $\operatorname{FID}$ of approximately $25$, consistent with the results reported in Song et al., 2019. Using our regularized objective with $r=10^{-4}$, we further improve the $\operatorname{FID}$ to $23$.
> Under a small time budget of $T=200$, the performance of the original score-matching method deteriorates significantly, yielding an $\operatorname{FID}$ of $160$, whereas our approach achieves an $\operatorname{FID}$ of $49$ with $r=5 \times 10^{-4}$. This demonstrates that our method consistently outperforms standard score matching, both under tight time budgets and for large $T$. We will include the corresponding results and generated samples in the main PDF.
>
> **Sample complexity**: We thank the reviewer for raising this valuable point. In fact, we already have sample complexity in our results (see Corollary 1). For further clarification, we will add a Remark with a detailed discussion to the paper on pages 6-7 in blue). A quick version is provided below:
>
> **Remark (sample complexity)**
> Corollary 1 states that once the network space is sufficiently large (that is, there exists a pair $(\Theta*, \kappa*) \in B_1 \times (0,\infty)$  such that $\kappa* s_{\Theta*}(x_t^i, t) = \nabla_{x_t} \log q_t(x_t^i)$ for all $i \in \{1,\dots,n\}$ and $t \in \{1,\dots,T\}$), the sample complexity of the regularized diffusion model increases by
> $$
> O (C_{x}^2 \frac{\log (nTp)}{\tau^2} \max ((s B)^4, (\kappa*)^4 ))
> $$
> in order to achieve
> $
> KL(q_0 , \hat{p}_0) \le \tau
> $.
>
>  This demonstrates a substantial improvement for our regularized estimator: the bounds now depend on the effective sparsity $s \ll d$ and grow only logarithmically with the total number of network parameters, rather than polynomially (compare with Block et al. (2020) and Gupta et al).
> More precisely, the bound can even decrease exponentially with the number of hidden layers of the network, thanks to our regularized objective, which we omit here for simplicity; see [Taheri2021] and Proof of Lemma 2 in our paper.
> In fact, regularization can reduce the effective complexity of the network space under consideration, thereby improving the sample complexity of score estimation. This is consistent with the results of [Zhu2023], who establish convergence rates of order $O(d\log \mathcal{N}(\tau,\mathcal{F})/\tau^2)$, where $\mathcal{N}(\cdot,\mathcal{F})$ quantifies the complexity of the network function class.
>
> **Employing other samplers:** Thank you for raising this question. Our analysis is built around a baseline sampler, and the proof of Theorem 1 relies on three dominant error sources—initialization error, reverse error, and estimation error. Since the reverse error depends on the effective dimension
>  s, the use of more advanced samplers
> [
> Benton et al., 2024; Gupta et al., 2024
> ]
>  may indeed improve the dependence on
> s. A detailed investigation of this potential improvement is an interesting direction for future work. In this paper, however, we keep the sampler fixed to the baseline in order to isolate and thoroughly study the impact of our regularization.
>
> **Regularization parameter r**:  As already mentioned in the paper, the tuning parameters are chosen as a decreasing function of T, say $r = f[T] \in O(c'/T)$ for some constant $c' \in (0,\infty)$. In almost all of our experiments, we used a tuning of order $1\mathrm{e}{-4}$  for large T and $1\mathrm{e}{-3}$ for small T (for CIFAR10, we used tuning $5e-4$ for small $T$ as the level of sparsity is lower than the MNIST family). Our empirical observations indicate that small changes in this order have only a minor impact on the results (results are stable for  $r \in [1e-4,5e-4)$). Moreover, the tuning needs to be selected only during the training phase---as diffusion models involve several hyperparameters to be set---while once the model is trained, it can be used for inference without further adjustment.
> Note that incorporating extra hyperparameters or architectural changes is common in improving generative models, and our tuning is no exception.
>
> We will update the PDF with the above-mentioned changes and the valuable references mentioned. We hope we can address the reviewers’ questions, and we would be happy to provide further clarification if needed.

---

### Author Response · Authors · 2025-11-24
**General response**

Dear All

We sincerely thank all the reviewers for their positive evaluation, insightful inputs, and questions. We have addressed each reviewer’s questions individually and will summarize below the main changes made in the updated version of our paper (highlighted in blue).

**New simulations with large model size (base channel 128) for CIFAR with FID metrics:**
We conducted a new simulation on CIFAR-10 using the standard model size (base channel 128). The original score-matching setup achieves an $\operatorname{FID}$ of approximately $25$, consistent with the results reported in Song et al., 2019. Using our regularized objective with $r=10^{-4}$, we further improve the $\operatorname{FID}$ to $23$.
For $T=500$, our regularized approach ends with $FID=25$, while original score matching drops to $38$.
Under a small time budget of $T=200$, the performance of the original score-matching method deteriorates significantly, yielding an $\operatorname{FID}$ of $160$, whereas our approach achieves an $\operatorname{FID}$ of $49$ with $r=5 \times 10^{-4}$. This demonstrates that our method consistently outperforms standard score matching, both under tight time budgets and for large $T$. We have now included the corresponding results and generated samples in the main PDF (Figure 6).

**Sample complexity**: We have added Remark 1 to clearly demonstrate the sample complexity implied by our results (see also the detailed discussion preceding Remark 1 for comparison).

**Run times**: We examined the training optimization time and convergence behavior for the regularized and original diffusion models on MNIST (refer to Section A7).

**Sample quality for toy data**: We evaluated the quality of the 2000 generated 3D samples (toy data) by comparing them to the true distribution using several metrics: Fréchet distance, MMD² with an RBF kernel, Energy distance, as well as error mean and covariances (see our response to the reviewer 5MyW).

We would be glad to clarify any questions that may remain.

---

### Author Response · Authors · 2025-12-02

Dear AC

We would like to summarise the changes made during the rebuttal period as follows:

1-In response to the reviewers (Hjd6;fKXu;5MyW), we conducted a new simulation with a large model size (base channel 128) for CIFAR10, reporting FID metrics: This experiment will demonstrate that our method consistently outperforms standard score matching, both under tight time budgets and for large $T$ (Section A.2; Figure 6). Corresponding reviewers confirmed  that this simulation 'could well answer their question' and that ``introducing the sparsity-inducing norm has observable effects on more complex datasets.'

2-Following the valuable input of the reviewer fKXu,   we examined the training optimization time and convergence behavior for the regularized and original diffusion models on MNIST (refer to Section A7). This simulation demonstrates that adding regularization has only a minor impact on training time, which could well address the reviewer's question.

3-Following the valuable input of the reviewer 5MyW, we evaluated the quality of the generated 3D-toy samples by comparing them to the true distribution using several metrics: Fréchet distance, MMD² with an RBF kernel, Energy distance, and mean and covariance errors. The reviewer explicitly confirmed  that ``my confusion has been well resolved, and I will maintain the positive rating.''

4-Following the valuable question raised by the reviewer La1N about the sparsity assumption, we referred to our detailed section A3,  showing that in the worst-case scenario, where the true score function is not sparse, our regularized estimator performs comparably to the original score-matching method. The reviewer mentioned that `the authors’ response has addressed my earlier concerns. I will keep my original score of 8'.

5-Following the valuable input of reviewer Hjd6, we added Remark 1 to clearly demonstrate the sample complexity implied by our results and a detailed discussion preceding Remark 1 for comparison with related works studying sample complexity.
The result demonstrates a substantial improvement for our regularized estimator: the bounds now depend on the effective sparsity and grow only logarithmically with the total number of network parameters. This result clearly shows that our method also nicely targets sample complexity. We did not receive any additional questions or inputs from the reviewer before the system process changed.

6-Following the valuable inputs by Reviewer fKXu, we analysed different hyperparameters, referring to simulations with other regularizations like $\ell_2$, elaborating about our motivation for regularization, and providing our source codes (that could well answer the reviewer's questions). We then discussed CFG, which is fundamentally different in nature from our approach as it is a conditional guidance mechanism (typically relying on prompts, conditioning models, ...), whereas our method operates in a purely unconditional setting.  And finally, we provided new simulation results on MNIST and CIFAR (score heatmaps and box plots), showing the sparsity in scores tabulated in our Section A.3 under "Sparsity within scores".  Our results clearly support our sparsity motivation and perspective. The corresponding reviewer mentioned explicitly their willingness to increase their score to 4,  and later to 6  by showing `Sparsity within scores', which is now added to the paper (under "Sparsity within scores" in Section A.3)

Thank you and best regards,

Authors

---

### Meta-Review · Area_Chair_xcNL · 2026-01-02

**Summary:**

The rebuttal improves the submission (added standard CIFAR-10 setting, additional quantitative evidence, training-time comparison, and code), but key risks remain. The core efficiency claim relies on sparsity/intrinsic-dimension assumptions that are not convincingly validated on more complex data, and the practical gains are demonstrated mainly under a weak/limited experimental regime (simple Langevin sampling, limited baselines, and small-scale datasets). The method is conceptually simple (regularization + scaling), and it is still unclear whether the reported speed–quality gains persist under stronger modern samplers/guidance baselines or larger-scale benchmarks. Overall, I recommend rejection due to insufficient empirical support and remaining uncertainty about the central assumption/mechanism.

**Reviewer Concerns:**

Addressed by rebuttal

- Added experiments in a more standard CIFAR-10 configuration and provided additional quantitative comparisons.

- Clarified sample-complexity statements and provided more implementation details (including code).

- Added analyses/visualizations related to score sparsity/magnitude and toy quantitative metrics.

- Provided training-time/optimization overhead comparisons and some guidance on hyperparameter scaling with T.

Still outstanding

- The central assumption/motivation (why true score fields should be sparse/approximately sparse in realistic settings) remains insufficiently justified; current evidence may reflect the effect of the regularizer rather than an inherent data property.

- Limited evaluation scope: no convincing evidence on larger/more challenging datasets and no demonstration that gains hold with stronger, commonly-used samplers and guidance baselines.

- Hyperparameter robustness is not fully established (sensitivity across datasets/architectures/time-steps), and the theory–implementation gap regarding constraint enforcement remains a concern.

- Novelty may be perceived as incremental without broader empirical validation or clearer mechanistic insight.

**Reviewer Scores:**

Hjd6: 4 -> 4/6 (some concerns mitigated by added standard CIFAR + quantitative/efficiency details, but still not fully convinced by assumption/baseline limitations).

fKXu: 2 -> 4 (substantial clarifications/added analysis, but remaining concerns about motivation/strong baselines).

La1N: 8 -> 8 (likely unchanged; already positive and rebuttal aligns with their expectations).

5MyW: 6 -> 6 (likely unchanged; rebuttal addresses several points but not enough to move strongly upward).

---

### Decision · Program_Chairs · 2026-01-26

Reject